

# Using MODIS estimates of fractional snow cover extent to improve streamflow forecasts in Interior Alaska

Katrina E. Bennett[1,2,3], Jessica E. Cherry[1,2,4], Ben Balk[5], Scott Lindsey[4]

5  [1]International Arctic Research Center, University of Alaska Fairbanks, Fairbanks, Alaska, 99775 U.S.A.
[2]Water and Environmental Research Center, University of Alaska Fairbanks, Fairbanks, Alaska, 99775 U.S.A.
[3]Current affiliation, Earth and Environmental Sciences, Los Alamos National Lab, Los Alamos, NM, 87545, U.S.A
[4]Alaska Pacific River Forecast Center, Anchorage, Alaska, 99502 U.S.A.
[5]AMEC Environment and Infrastructure, Boulder, Colorado, 80302 U.S.A.

*Correspondence to*: Katrina E. Bennett, kbennett@lanl.gov, 505-664-0698, Earth and Environmental Sciences, Los Alamos National Lab, Los Alamos, NM, 87545, U.S.A





**Abstract** Remotely sensed snow cover observations provide an opportunity to improve operational snowmelt and streamflow forecasting in remote regions. This is particularly true in Alaska, where remote basins and a spatially and temporally sparse gaging network plague efforts to understand and forecast the hydrology of subarctic boreal watersheds and where climate change is leading to rapid shifts in watershed function. In this study, the operational framework employed by the US National Weather Service, including the Alaska Pacific River Forecast Center, is adapted to integrate Moderate Resolution Imaging Spectroradiometer (MODIS) remotely sensed observations of snow cover extent (SCE) to determine if these data improve streamflow forecasts in Interior Alaskan river basins. Two versions of MODIS fractional SCE are tested in this study: the MODIS 10A1 (MOD10A1), and the MODIS Snow Cover Area and Grain size (MODSCAG) product. Observed runoff is compared to simulated runoff to calibrate both iterations of the model. MODIS-forced runs have improved snow depletion timing compared with snow telemetry sites in the basins, with discernable increases in skill for the streamflow simulations. The MODSCAG SCE version provides moderate increases in skill, but is similar to the MOD10A1 results in these watersheds. The basins with the greatest improvement in streamflow simulations have the sparsest streamflow observations. Considering the numerous low-quality gages (discontinuous, short, or unreliable) and ungaged systems throughout the high latitude regions of the globe, this result is of great value and indicates the utility of the MODIS SCE data in these regions. Additionally, while improvements in predicted discharge values are subtle, the snow model better represents the physical conditions of the snow pack and therefore provides more robust simulations, which are consistent with the US National Weather Service's move toward a physically-based National Water Model. Physically-based models may be more capable of adapting to changing climates than statistical models tuned to past regimes. This work provides direction for both the Alaska Pacific River Forecast Center and other forecast centers across the US to implement remote sensing observations within their operational framework, to refine the representation of snow, and to improve streamflow forecasting skill in basins with few or poor-quality observations.

## 1 Introduction

Arctic climate change is rapidly transforming the North with myriad impacts on the hydrologic realm, which has major implications for the largest biome on earth, the boreal forest. For the northernmost US state, Alaska, climate change has affected the hydrology, ecology, and society in significant ways (Euskirchen et al., 2009, Hinzman et al. 2005, Hinzman et al. 2013, Wrona et al. 2016). Alaska has warmed more than two times the rate of the rest of the US since the 1950s (Karl et al., 2009). Interior boreal Alaska has warmed the most of all regions in the state, increasing by 4°C in winter and 1.9°C annually from 1949-2011 (Stewart et al., 2013). Snowpack extents and duration in Alaska have decreased over time by 18% (1966-2012) due to an earlier snow melt (SWIPA, 2012). Changes in temperature and snow are also affecting frozen ground and leading to decreases in the permafrost—the temperature of the permafrost near Fairbanks Alaska has risen by 2-4°C from 1930-2003 (Slater and Lawrence, 2013; Koven et al., 2015). Rivers in Alaska have been observed to be changing as a result of an intensified or stronger hydrologic cycle that could lead to an increase in peak flows in the Northern American high latitudes (Huntington, 2006; Rawlins et al., 2010). The intensification is owing to the Clausius–Clapyeron relation that dictates an exponential increase in





specific humidity with increased temperature (Huntington, 2006; Cohen et al. 2012). Riverine breakup dates have been noted to be occurring earlier (Cooley and Pavelsky, 2016; Lesack et al. 2014; Muhammed et al. 2016). Extremes events are also changing; annual maximum streamflow trends indicate that Alaskan riverine systems are experiencing streamflow declines, while minimum flow trends are largely increasing (Bennett et al. 2015). All of

5 these shifts are leading to increased streamflow variability (Stuefer et al. 2017), which has strong impacts on the infrastructure and economy of Alaska, and the Arctic as a whole (Instanes et al. 2016), leading to a massive challenge in terms of observing, understanding, mitigating and adapting to these effects. The Far North (Arctic and Subarctic) is also rapidly developing its hydroelectric water resources, unlike the contiguous US, and needs accurate decision support for managing this infrastructure (Cherry et al., 2017; Sturm et al. 2017).

An enormous challenge for scientists attempting to accurately represent the impacts of climate change on the Alaskan hydrosphere is the vast territory, complex landscape and sparse observational network. Alaskan hydrologic systems suffer from large uncertainties in various data inputs, and thus require great care when attempting to simulate hydrologic water balance components with skill. For example, precipitation measurements are of very poor quality in winter (Cherry et al. 2005; 2007; Groisman et al. 2014) and river stage and discharge measurements by automated

gages do not read accurately when ice is present in the river. Reducing these uncertainties is of utmost importance, as they will reduce the value of model output model output (Magnusson et al., 2015; Slater et al., 2013; Clark et al. 2017) and the results cannot provide actionable guidance on water resource management (Stocker et al., 2013). In addition, the variability in landscape (i.e. forest cover, topography, discontinuous permafrost) and climate across Alaska require robust modeling techniques to account for potential climate-driven shifts. This adaptable approach is

increasingly important as the NOAA's National Weather Service (NWS) develops the National Water Model (NWM) framework, a multi-scale water prediction model in operations over the contiguous US (NOAA, 2017). Temperature index models, based on the most reliable climate forcing, are often presumed to perform better in regions with highly variable landscapes and a sparse network (Hock, 2003; Stahl et al., 2006). Alternatively, a skillfully calibrated conceptual model may provide a better representation of hydrologic responses because the

underlying model is reliant upon parameterizations rather than observations that lack spatial and temporal consistency (Franz et al., 2008; Reed et al., 2004).

To deal with the inoperability of stream gages during breakup and *in situ* snow observations, one technique is to use remotely sensed snow cover areal extent (SCE) to supplement point observations such as temperature, precipitation and streamflow commonly used both as model inputs and for model calibration and validation (Parajka and Blöschl,

2008). There are two main ways that this data has been used to date: either to directly insert a time series of SCE data into the model (McGuire et al., 2006; Rodell et al., 2004), or to use complex assimilation procedures to filter the snow series and merge it with observational data (Andreadis and Lettenmaier, 2006; Sun et al., 2004; Zaitchik and Rodell). There is a concern that direct insertion methods are ineffective at improving streamflow models and do not perform better than uninformed models because melt can occur before snow cover drops below 100% (Clark et al.,

2006). In addition, the melt season duration is often short, transitioning rapidly from snow-covered to snow-free, although this is largely basin-dependent (Clark et al., 2006). Assimilation approaches have yet to be integrated into operational models, in part because of the limited research showing the impacts of assimilation on the hydrologic



forecast. Other studies have found calibrating models based solely on SCE values may not improve skill in estimating discharge, and the improvements for in-catchment distributed SCE estimates do not always result in improved discharge simulation (Franz and Karsten 2013; Duethmann et al., 2014). However, Liu et al. (2013) and Thirel et al. (2013) found marked improvements in land surface model output for basins in Alaska where data

assimilation processes were applied.

One approach to improve streamflow forecasts under climate change is to utilize newly developed frameworks to ingest remotely sensed data on snow cover extent into streamflow models. These newer tools have been adopted by the NWS's River Forecast Centers (RFCs) and offer an opportunity for more advanced streamflow forecasting techniques, including ensemble prediction using variable input and/or forcing data. The Community Hydrologic

Prediction System (CHPS), brought online in 2012 by the Alaska Pacific River Forecast Center (APRFC), is a test case for this approach. The modeling framework, developed on the Delft-FEWS software platform, can run many different types of models, but in its current state implements the conceptual Sacramento Soil Moisture Accounting System (SAC-SMA) rainfall-runoff model (Burnash et al., 1973), with snowpack input from the SNOW17 snow model (Anderson, 2006).

The objective of this paper is to adapt the CHPS operational forecasting modeling framework to ingest Moderate Resolution Imaging Spectroradiometer (MODIS) remotely sensed SCE data for improved streamflow modeling of the Interior boreal forest region of Alaska within sparsely and poorly-observed river basins that are experiencing shifts associated with a changing climate. We replace the standard areal depletion curve used in SNOW17 with pre-processed MODIS SCE grids for snow depletion. Two different versions of MODIS are applied: the MOD10A1

fractional SCE product, which is the standard MODIS global snow cover product (Hall et al., 2002), and the MOD-Snow Covered Area and Grain size (MODSCAG) fractional SCE product, which is a regional product (Painter et al., 2009). The SNOW17 manual calibration using all model parameters is evaluated, including a tolerance parameter controlling snow cover updates (snow cover tolerance, SCTOL), to simulate a mixed method between direct insertion and more complex data assimilation. Pre-processing, model frameworks and use of existing parameterizations are

thus offered as a means of incorporating remotely sensed information into operational models that can be utilized out-of-the box by the NWS RFCs. The paper also examines issues around the use of MODIS SCE in high latitude boreal forest basins, the interpolation of missing data, and the improvement of streamflow estimates by calibrating model parameters used in streamflow forecasting systems across the US.

## 2 Methods

### 2.1 Study area

This study was carried out in five adjoining headwater sub-basins of the Tanana River, which is a sub-basin of the Yukon River basin (Figure 1). The sub-basins include the Chatanika, Upper Chena, Little Chena, Salcha, and Goodpaster basins. The Chatanika River basin above the Steese Highway (64°50′37″N, 147°43′23″W; Figure 1) is approximately 950 km$^2$ in size and is oriented predominantly east to west. Only the area upstream of the Caribou-

Poker Creek confluence is considered in this study. The Chatanika was gaged from 1987 to 2007 but the records are




highly discontinuous. The Upper Chena River basin is approximately 2440 km$^2$ and has gage records from 1967 to present. This portion of the basin contains high elevation peaks and rocky outcrops where snow can persist late into the melt season. The Little Chena is 1030 km$^2$ and contains the highest proportion of lowlands relative to the other basins; it has been gaged since 1966 to present. The Salcha River watershed is a large, 5740 km$^2$ basin with its gage

at the Salchaket Bridge and has the longest historical record of all rivers in this region (1948 to present). The Goodpaster basin is located east of the Salcha and is 1770 km$^2$ in size. It has the highest proportion of its basin above 600 m elevation and has been gaged since 1997 to present. Upper watersheds are split into sub-basin units with north and south facing aspects, with the exception of the Little Chena. There are minor urban and agriculture developments throughout the region, including the town of Fairbanks, which is located downstream of the Little Chena gage on the

main stem of the Chena River. These minor developments have little or no bearing on the hydrologic response of the headwater systems of Chena basins we examine here. More information on the watersheds is provided in Table 1.

## 2.2 Data

The MODIS satellite product (Terra MOD10A1, version 5) provides daily, 500 m resolution snow cover fractional areal extent (SCE) data. It was downloaded from the National Snow and Ice Data Center (Hall and Riggs, 2007; Hall

et al., 2006; Riggs et al. 2006) for 2000-2010 and pre-processed into projected GeoTIFFs (North Pole Stereographic). MODSCAG data products were obtained from the NASA Jet Propulsion Laboratory's Snow Data System Portal (http://snow.jpl.nasa.gov/) for the area of interest and pre-processed into projected GeoTIFFs to match the spatial properties of the MOD10A1 data. Further information on pre-processing requirements and adjustments for both MODIS data products are provided in the supplemental materials (Supplement, Sect. 1.1).

Mean areal values of temperature and precipitation at 6-hour increments are obtained for each sub-watershed from the APRFC for the time period 1969 to 2012; only the 1999-2010 data are utilized in this study. River discharge at each gage is based on the US Geological Survey (USGS) gaging record database. The exception to this is the Chatanika River at the Steese Highway site, where observed discharge is generated based on once-a-day stage readings from a Cooperative Network observer. These daily stage readings are converted to mean daily discharge

using the APRFC's rating curve for the river. Aspect and elevation were calculated using the 30 m US Geological Survey's National Elevation Dataset (NED), updated for the region in 2012 (Gesch et al., 2002). Seven snow telemetry (SNOTEL) sites are utilized to compare simulated SWE with observed data (Table 2, NRCS 2013). SNOTEL snow water equivalent (SWE, mm) is downloaded from the National Resource Conservation Service (NRCS) snow pillow data repository (http://www.wcc.nrcs.usda.gov/ftpref/data/snow/snotel/cards/alaska/).

Potential evapotranspiration (PET) estimates are provided by the APRFC based on an assessment of historical potential evapotranspiration from pan evaporation data and Thornthwaite estimates (Anderson 2006). These data are used to develop a general linear relationship between PET and elevation to estimate average monthly PET values for a generic low elevation site. The APRFC uses the low elevation PET values to derive monthly estimates for the mean elevation of each sub-basin as a coefficient. The coefficient, C, is derived using the equation,

$$C = 0.9 - [(e-1000) \cdot 0.00011]$$

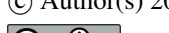



where $e$ represents elevation in feet. For example, if the catchment mean elevation is 716 m, the coefficient is 0.75. Finally, a monthly PET adjustment factor is applied to account for vegetation changes during the year. The result is an evapotranspiration demand estimate that is used in the SAC-SMA model, described in the next section.

### 2.3 Models

The SNOW17 and the SAC-SMA models are run by the APRFC in an operational framework referred to as CHPS. CHPS is built upon the Delft Flood Early Warning System (FEWS), developed by Deltares. The CHPS system is briefly described in the Supplemental materials section (Supplement, Sect. 1.2).

### 2.3.1   SNOW17

The SNOW17 snow model is a single layer snow model that calculates snow accumulation and ablation using
empirical formulae to estimate heat and liquid water storage, liquid water throughflow and snowmelt (Anderson, 1976). The model is designed for river forecasting and has been used operationally by the NWS RFCs since the mid-1970s. The only input requirements for SNOW17 are temperature and precipitation (winds are accounted for but not input as observations), at the model time step (6 hrs). There are 12 parameters in the SNOW17 model, including the areal snow depletion curve; sensitive or 'major' parameters control the model outputs while less sensitive or 'minor'
parameters have little impact on the model output (Table 3, He et al., 2011).
SNOW17 determines the division between rain and snow using the rain-snow elevation (RSNWELEV) module. RNSWELEV uses a defined lapse rate (6ºC/100m) to determine the air temperature threshold that results in rain turning to snow (PXTEMP, Table 3). This temperature threshold is related to an elevation and is passed to SNOW17, the percent area above and below that elevation is determined from a defined area elevation curve. Multiplying these
percentages by the precipitation thus defines the proportion of precipitation falling as snow or rain in the basin. Non-rain snowmelt (mm) is determined from air temperature minus the baseline temperature at which melt occurs (MBASE; set to 0ºC), weighted by a seasonably variable melt factor that is calculated using an oscillating sine curve that varies between the minimum (MFMIN) and maximum (MFMAX) melt factors for December 21$^{st}$ and Jun 21$^{st}$ (mm/ºC/6 hrs). These values are adjusted for latitudes above 54ºN to account for low radiation input, a paucity of
25   days when temperatures rise above freezing, and rapid changes in melt rates during spring and fall (Anderson, 2006). A fixed lapse rate is applied to mean air temperature within the lumped basins for the elevation at which the air temperature time series is collected (TAELEV), in the case when TAELEV differs from basin mean elevation. This fixed lapse rate can be configured in the SNOW17 model using parameters that define the lapse rate at time of maximum/minimum temperature, and is set to 0.6ºC/100m in this study.
A simplified energy balance method is used to calculate melt from rain-on-snow, based on the following assumptions: (1) use of the Stefan-Boltzmann constant for incoming longwave radiation; (2) negligible shortwave radiation; (3) 90% relative humidity; and (4) accounting for wind speed by adjusting for the average value of the wind (mm/mb/6hr) during rain-on-snow events (UADJ). Heat content within the snowpack is calculated based on a gradient difference between air temperature and the near-surface snow pack temperature index to determine the heat
flow direction when melt is not occurring. Depending on the near-surface snow pack temperature index, more or less





weight is assigned to temperatures from previous time intervals to represent deeper or shallower snow pack temperatures.

The snow heat deficit is either negative or positive; the rate of heat loss or gain is based on the amount of energy exchange that occurs when melt is not taking place at the snow surface (defined as the negative melt factor, NMF),

which is weighted by MFMAX to account for seasonal variations in pack heat translation. Heat can also be translated from the ground to the snow using a parameter that controls the daily melt volume at the interface between snow and soil and is assumed to occur continuously through the snow season (DAYGM). When the snowpack is at peak water-holding capacity (PLWHC) and is isothermal at 0ºC, the snow is ripe and any excess water entering the snow will flow through it as outflow. Water movement through a ripe pack is attenuated or lagged based on empirical formula

derived from lysimeter studies (Anderson 2006).

### 2.3.2 SCE in SNOW17

SNOW17 uses an areal depletion curve (ADC) to represent the snow cover extent; the ADC is used to calculate the area of the basin over which surface melt, changes in heat storage, ground melt, and rainfall on bare ground occurs. The ADC not only represents areal extent of snow cover, but also accounts for slope, aspect and differences in

vegetative cover (i.e. open versus closed sites). In the baseline model run, the areal extent of snow cover was calculated from a lookup table that defines the ADC and relates it to the ratio of SWE to either a) the maximum value of SWE that occurred during snow accumulation or b) a parameter (SI) that represents the areal SWE at which 100% snow cover exists (referred to as the areal index). The ADC in the baseline model run is applied as follows: when snow accumulates, the snow cover is set to 100%, and it stays at this value until it falls below SI or the maximum

SWE value, whichever is smaller. If new snow totaling greater than 0.2 mm/hr falls onto bare ground, 100% snow cover is assumed until 25% of the new snow has melted. For Alaska, several different ADC configurations are used depending on whether slopes are south versus north facing, or in upper versus lower elevation basins. The watersheds in this study used the same ADC for upper south, upper north and lower sub-basin units since they have similar orientations in a similar geographic region. Only the Little Chena uses a different ADC for its upper watershed, as no

north/south aspect split is used in this basin. For all other model runs, the ADC was replaced by areal extent of snow cover derived from the two MODIS SCE datasets (Figure 2). Other parameter settings used to alter the impact of the MODIS SCE data in SNOW17 are described in the Supplemental, Section 1.3.

### 2.3.3 SAC-SMA

The SAC-SMA model is a conceptual rainfall-runoff model that produces streamflow simulations from observed

input precipitation and PET (Burnash et al., 1973). SAC-SMA has been widely applied by the NWS to estimate streamflow runoff in basins across the US. The model moves water into either an upper or lower storage zone that conceptually represent soil interception or deep groundwater storage. Interception water in the upper zone flows to the lower zones via downward percolation or can run off directly or via interflow when the upper zone layers become saturated and the precipitation rate exceeds downward percolation. Lower zone water can be held in tension storage

and contribute to baseflow runoff slowly over time or can run off more quickly over shorter durations. Drainage from



the upper and lower zones follows gravity drainage and is governed in part by both water delivery from the upper zone and soil moisture in the lower zone. Tension water is driven by potential evapotranspiration and diffusion, with a fraction of the lower zone unavailable for potential evapotranspiration as it is considered below the rooting zone. A unit hydrograph model is used to adjust runoff timing for each lumped watershed in the SAC-SMA model. Each

sub-watershed has its own unit hydrograph to translate the runoff through the channel system to the gage location. Simple routines sum the unit hydrograph outputs to calculate simulated streamflow at the basin outlet. While downstream basins incorporate routing models to move water from upstream to downstream basins, this study focuses on headwater basins so no routing models are needed.

## 2.4    Calibration

Several calibration procedures were undertaken for this project; the baseline calibration, and the two MODIS data set calibrations. The baseline calibration effort updated the SAC-SMA/SNOW17 model parameters to the 2000-2010 study years used in this study, as they had previously been adjusted by APRFC to 1970-2003 historical data. The two MODIS manual calibrations used the updated baseline to adjust parameters and generate statistics. Calibration entailed using both visualizations of streamflow hydrographs from 2006-2010 and statistics from the entire period of

record for ultimate parameter selection.

To calibrate the MODIS model output, a simple approach is taken to minimize the terms required for calibration. This ensures that it was a) easy to replicate the model adjustments to the MODIS SCE data and b) solely focused on the snow parameterization, as adjustments to the SAC-SMA parameters resulted in only minor improvements to model calibration statistics during the spring ice breakup period. Also, priority was placed on adjusting the empirical

parameters towards a physically-based realization using watershed and sub-basin unit properties, including the topographic aspects and the observed melt trajectory impacted by the MODIS SCE data. To complete this simple, physically realistic calibration approach only the parameters MFMAX and TAELEV were adjusted. Further details of the calibration efforts are described in the Supplemental, Section 1.4.

## 2.5    Validation

For validation purposes, statistics from 2000-2005 are provided for all watersheds except the Chatanika. The Chatanika basin was calibrated using 2000-2004 data and validated from 2005-2010 to make use of the better data quality and availability during the first five years of the study. Statistics used to evaluate model success are based on five main objective functions. The first two of these criteria are standard in NWS RFC calibration approaches and are provided in the CHPS statistical output. These statistics were used for evaluation during the calibration; total volume

bias as a percent (PBIAS, %) and the correlation coefficient (R, unitless). An additional three objectives were added for further validation of the results, Nash Sutcliffe efficiency (NSE, unitless), the mean absolute error (MAE, $m^3$/sec) and the root mean squared error (RMSE, $m^3$/sec). Statistics were run only for April, May and June to focus on the changes to the snowmelt season; March is not included because generally, river ice melts and breaks up in Interior Alaska in March, thus any differences in statistics would be indicative of changing winter conditions rather than

changes in spring snowmelt timing or volume. The equations are calculated as follows:





$$PBIAS=\left[\frac{\sum_{i=1}^{N}(S_i\text{-}Q_i)}{\sum_{i=1}^{N}Q_i}\right]*100 \qquad (2\text{-}1)$$

$$R=\frac{N\cdot\sum_{i=1}^{N}S_i\cdot Q_i-\sum_{i=1}^{N}S_i\cdot\sum_{i=1}^{N}Q_i}{\left[\left(N\cdot\sum_{i=1}^{N}S_i^2-\left(\sum_{i=1}^{N}S_i\right)^2\right)\left(N\cdot\sum_{i=1}^{N}Q_i^2-\left(\sum_{i=1}^{N}Q_i\right)^2\right)\right]^{0.5}} \qquad (2\text{-}2)$$

$$NSE=1-\left(\frac{\sum_{i=1}^{N}(S_i\text{-}Q_i)^2}{\sum_{i=1}^{N}(\bar{Q}\text{-}Q_i)^2}\right)$$

$$MAE=\sum_{i=1}^{N}\left|S_i\text{-}Q_i\right|$$

$$RMSE=\left[\frac{\sum_{i=1}^{N}(S_i\text{-}Q_i)^2}{N}\right]^{0.5}$$

where N is equal to the number of data points (i.e. sub-daily streamflow realizations), i is the time step (days), S is

10 the simulated streamflow ($m^3$/s), and Q is the observed streamflow ($m^2$/sec).

## 3 Results

### 3.1 Baseline Model Results

The APRFC SAC-SMA/SNOW17 baseline model estimates of streamflow in Interior Alaskan river basins for the 11-year period of record indicate that these watersheds are captured with skill (Table 4). The Chatanika basin is

15 problematic given the limited quality and quantity of the observed streamflow data, as noted in the statistics below for each objective function. For all of the five basins analyzed, the daily average bias for the period of record is ±3% or less. Daily correlation coefficients (R, unitless) are equal to or greater than 0.84 and higher for the four watersheds with quality observed data, while the Chatanika basin is 0.70. NSE (unitless) daily values are also above 0.60 for all basins except the Chatanika, which is 0.18 due to the noise in the observed data values. Daily mean absolute error

statistics are below 10 $m^3$/sec for all basins except the Salcha, which is 15.89 $m^3$/s owing to its long discharge record. RMSE ranges from 3.5 $m^3$/s (Chatanika) to 33 $m^3$/s (Salcha). Across all basins, SCE is variable by elevation zones and years (Figure 3). Upper elevation areas tend to have 100% SCE, while mid-to-lower areas often begin the year with 75% SCE or less. The very lowest elevation zone appears to have a slightly higher SCE values than two adjacent higher elevation zones (Figure 3). Some years have a markedly late melt out, with high variability across all

elevation bins. Lower elevation zones tend to melt out in early April, while the upper regions of the watersheds hold snowpack weeks or months into the subarctic spring (Figure 3).

### 3.2 SAC-SMA Model MODIS Calibrations

Calibrated SNOW17 parameters for the APRFC and MOD10A1 runs resulted in increased MFMAX for north facing aspect in two sub-basin units and increased TAELEV for the northern slopes (Table 5) compared to the baseline

APRFC SAC-SMA/SNOW17 run. In some sub-basin units, TAELEV was set to be equal for the north and south



slopes, for reasons that are discussed in the following section. MFMAX for the Chatanika's lowland sub-basin increased and TAELEV at the north sub-basin was increased, while TAELEV was decreased for the south sub-basin unit. MFMAX in the Upper Chena north was unchanged and TAELEV was equalized for both south and north sub-basin units. The Little Chena sub-basin parameters were altered by setting MFMAX equal to its maximum

recommended value (1.4) for the upper and lower sub-basins and by increasing TAELEV 100 m greater than the elevation for both sub-basins. TAELEV for Salcha and Goodpaster were differenced by 100 m for the north and south sub-basin units and the northern sub-basin MFMAX for Goodpaster was increased slightly. Goodpaster's lower basin MFMAX was reduced by a small amount. Although these changes may appear minor, MFMAX is highly sensitive during the melt season and therefore these changes have a substantial effect on the MODIS SCE forced

snowmelt trajectory at these sites (Anderson, 2006).

In the MODSCAG runs, values for MFMAX were increased slightly for the north sub-basin units for all basins. TAELEV values were adjusted slightly in Upper Chena, Salcha and Little Chena bains (Table 6), but were not altered from the baseline run in Chatanika. In the Goodpaster basin, the TAELEV value for the south sub-basin unit was decreased. For this version of MODIS, it appears that a slightly more rigorous calibration was required.

NMF was altered slightly for both MODIS runs to account for different snow densities and thermal conductivities of snow on south and lowland sites versus north aspects. Snow density is generally low in Interior Alaskan watersheds; based on analysis of field data from the Caribou Poker Creek watershed, snow density on the sites is approximately 0.20 and is slightly higher on the southern sites compared to the north site. The northern facing slopes were therefore given the NMF value of 0.15 mm/°C/6hr, which Anderson (2002) indicates is a 'reasonable' value of NMF. The

south and lowland sites, which have generally warmer temperatures and more dense snow, were assigned the NMF value of 0.2. For these runs, SCTOL is set to 0 for all basins to ensure that the MODIS data are utilized 100% of the time.

### 3.3   SCE and SWE

Compared to the APRFC runs, the MODIS runs have less snow cover on the north facing slopes and more on the

25 south facing slopes (Figure 4, Upper Chena River basin results for 2001 are shown as an example). Differences between the two runs become discernable in late January as a result of the different calibrations of the SNOW17 model in the watersheds (Figure 4), with larger differences at the north sub-basin units compared to the south sub-basin unit. As soon as the MOD10A1 SCE begins to alter the weighting factors for outflow from the snow, differences between the SWE generated by APRFC and MODIS runs are observed. The greatest differences between

30 the model runs occur during the melt season. All model runs peak in early April and start a downward melt trajectory, reflecting melt patterns at the upper elevation SNOTEL sites: Mt. Ryan, Munson, and Upper Chena. The APRFC and MOD10A1 run melt out later than the MODSCAG SCE north unit and the MODSCAGE estimates are closer to the APRFC runs in volume, although all runs terminate on the same approximate day for the northern sub-basins.

The SNOTEL sites are mostly located at upper elevations (Mt. Ryan, 850m and Munson, 940 m) compared to the SNOW17s ~800 m elevation parameter and thus illustrate conditions exhibited at high elevation northern sites in the



watershed. Mt. Ryan, in particular, does not build a snow pack early in the season, perhaps owing to its open, mountainous and presumably windy environment. The SNOW17 model is run over a lumped area so there is mix of site conditions that act to smooth the model responses; hence the comparison between SNOTEL SWE and SNOW17 modeled SWE are inherently qualitative as opposed to quantitative (Molotch and Bales, 2005). The lower elevation

SNOTEL sites, Teuchet and Little Chena, show earlier melt out than is seen in either the model output or the MODIS datasets. There is stronger coherence in the response of the northern sites as opposed to the southern sites. In the south sub-basin units, the MODIS runs melt out later, with MODSCAG again having the latest melt, similar in timing to the high elevation stations.

The areal extent of snow cover varies across the watersheds in both runs. The preprocessed gridded MOD10A1 SCE

illustrated for May 15th, 2001 for the watersheds is shown in Figure 5a and the MODSCAG SCE is show in Figure 5b. The high elevation snow pack (blue) is present within the upper watershed regions but the pack is largely gone in the valleys and lower watershed reaches. This translates into the lumped average SCE estimates shown in Figures 5c and 5d, which illustrate how CHPS ingests and converts the gridded MODIS SCE for the watersheds sub-basin units. North and south sub-basin units are differentiated in the upper sub-basin units (see Table 1) but not at other locations

because both aspects have begun to melt by this date (as opposed to early in the melt period when the south slopes would have comparatively less SCE than the north slopes). MODSCAG has less cloud cover interaction in this scene (Figure 5b) and this results in slightly higher values of SCE (Figure 5d).

SWE estimates for MOD10A1 (Figure 6a), MODSCAG (Figure 6b), and the difference between the MODIS (both versions) and APRFC run (Figure 6c and 6d) is shown for May 15th, 2001. Sub-basin units can be clearly

differentiated in these plots, which illustrate the range of SWE values from 0-0.5 inches in the lowland regions to 5 inches remaining in the upper headwaters. The MODSCAG data has an average SCE value of 0.5 and SWE is 1.7 inches, whereas the MOD10A1 has an average of 0.45 SCE, an average of 2.1 inches SWE, very small differences overall although watershed-to-watershed and sub-basin-to-sub-basin the variation between the products is notable. The difference plots highlight the fact that MODIS tends to have lower SWE values compared to the APRFC

SNOW17 model runs on the north facing slopes and higher values on the south facing slopes. The APRFC tends to be have lower SWE estimates for the lowland regions, although this is more true for MOD10A1 than MODSCAG (Figure 5c, d).

### 3.4    Streamflow Estimates

Calibration and validation results are provided for April-May-June (Table 4) for the MODIS and APRFC runs. For

MODIS data, many statistics are similar or nearly identical to the APRFC run with slight declines in model performance and some gains (Chatanika, Little Chena), particularly for the analysis focused on the whole period of record (Table 4). NSE statistics are particularly poor for all runs in the Chatanika basin, where the lack of continuous and high-quality observations hamper calibration efforts. The MOD10A1 data improves streamflow simulations in the Chatanika and Goodpaster systems during the calibration period, while it performs similarly or slightly worse

during the validation and period of record in most of the watersheds except the Chatanika. The MODSCAG run exhibits better performance compared to the APRFC run during the calibration periods in the Chatanika, Salcha and



Goodpaster basins, while the validation period statistics showed improvement for the Chatanika, Little Chena, and Upper Chena basins. Overall, the greatest improvements in skill are observed for the MODIS runs in the Chatanika and Goodpaster basins, the validation period for Upper Chena and the calibration period for Goodpaster (Table 4). Figure 7 shows the calibration, validation and whole period of record results are provided for all watersheds for R-1

plotted against RMSE. In the poorly performing watersheds, MODSCAG (and MODSCAG with SCTOL=0.25) tends to do slightly better versus APRFC in the calibration/validation time where improvements are also made for MOD10A1, while both MODIS versions perform nearly identically over the 11-year period. This can also be observed from the analysis presented in Figure 8 for all five watersheds. Here the percent differences from the observed are plotted as the APRFC differences against the MOD10A1 and MODSCAG products for comparison

between the three estimates. The plots illustrate that the MODSCAG results tend to follow more closely (and are hence more constrained) with the APRFC results, while the MOD10A1 product has more scatter. However, the differences from observed are similar between the two products.

Average (2000-2011) streamflow for each basin shown in Figure 9 highlights variations between simulated discharges plotted against observed discharge at the streamflow gages. Plots illustrate the average of all years in each

panel, with average of the five watersheds provided in the last panel. Only March to June results are shown; in March the watersheds have not begun to melt and the hydrograph depicts baseflow contributions in the systems. The active period begins in late March to early April and the differences between the two estimates of streamflow persist until June, after which point streamflow responses to rainfall input are essentially the same. Statistics for the April-May-June calibration, validation and the period of record are also provided in Table 4. The Upper Chena River basin

shows improvement compared to the APRFC run during the early melt period, while the later period is over predicted by the MODSCAG. For Chatanika, the simulated MODIS runs are of greater magnitude (Figure 9) and have earlier timing compared to the APRFC simulated flows. In the Little Chena river basin, MODIS simulated discharge overall fits better than the APRFC, which over simulates streamflow on average; and both products perform similarly well. Streamflow simulations for the Upper Chena, Salcha and Goodpaster systems on average

match observed more closely by the MODSCAG runs. This also is clear from the averages across watersheds and years; the MODSCAG simulations match observed streamflow, while the MOD10A1 product underestimates runoff during the mid-May to early June period (Figure 9, last panel).

### 3.5    Other Integration Methods

Two methods were applied to integrate the MODIS data into CHPS. One method involved interpolating between

missing data values, changing the number of interpolated days from 1 to 11 to investigate how changing the value impacted model results. Generally, the number of days of interpolation had little impact, but the longer interpolation period results produced slightly higher correlations and improved streamflow estimation. We also investigated the response to altering model parameter SCTOL, which can be used by forecasters to combine the strength of the ADC and the MODIS data and is similar to partial rule-based direct insertion approach, however the parameter can be

altered without any additional changes to the CHPS model framework. Because this. Table 7 illustrates the results of setting the SCTOL parameter to 0.25, 0.50, and 0.75 for the MODSCAG run only, while holding the rest of the



parameters constant. No recalibration is performed. NSE and R statistics increase during the calibration period, MAE and RMSE remain similar on average but the range of responses across the basins decreases for SCTOL=0.50. Interestingly, Chatanika, which has the greatest improvement based on the differences between APRFC and MODIS runs (Chatanika) does not benefit from model integration, owing to the low skill within the APRFC model version

(Table 7). However, for the remaining basins strong improvements are apparent for higher values of SCTOL during the calibration period (Upper Chena, Little Chena and Salcha), validation and period of record (Upper Chena, Little Chena). Diminishing returns occur at a threshold between 0.25 and 0.50 SCTOL for most basins; however, Goodpaster improves at 0.50 but not 0.75. This suggests that the SCTOL parameter should be uniquely applied dependent upon the basin.

**4    Discussion**

Results illustrate that streamflow in interior Alaska can be simulated with skill using conceptual, semi-lumped hydrologic models, even without the use of gridded observations of MODIS SCE. However, if the initial streamflow observations are of poor-quality (i.e. Chatanika River basin), applying gridded observations of MODIS SCE in the models will generate streamflow estimates as good as or better than estimates based on SNOW17s areal depletion

curve. However, as the climate shifts, conceptual, semi-lumped models may not be representative of process changes that will likely occur as the Arctic warms (Clark et al. 2017). As fully process-based models are challenging to run in Arctic environments, where high quality data is temporally and spatially sparse, using conceptual models parameterized with as many observations as possible represents a bridge between the fully processed based models and conceptual approaches to hydrologic modeling.

However, we found there to be major challenges in obtaining improvements in simulated streamflow discharge values when introducing additional observed data sets and their associated uncertainties into models. This result was also found in work performed in the American River basin where the California Nevada RFC lumped model provided the most accurate representation of snow cover area (Franz and Karsten 2013). As indicated by Franz and Karsten (2013), although the gridded representation of SCE is improved in their distributed version of SNOW17, the

streamflow simulations and associated statistics did not reflect this improvement. In addition, they found that discharge values had lower skill when estimates of snow cover are included in the calibration even though it is hypothesized that the process representation is improved, which is a finding of a number of other research studies focusing on this topic (Parajka and Blöschl, 2008; Udnæs et al., 2007). These findings are also true for Alaskan interior boreal watersheds, highlighting the importance of performing this work in remote and under monitored

systems that are changing quickly due to climate shifts and increased occurrences of extreme events (Bennett and Walsh, 2015; Bennett et al. 2015).

The goal of this work was, in part, to undertake a simple application of inserting preprocessed MODIS SCE into the CHPS operational framework to simulate streamflow across basins in Interior Alaska. The preprocessing of MODIS data for insertion into the model, which included the MOD10A1 and MODSCAG data products, along with the

CHPS areal averaging eliminated some of the issues related to cloud cover and missing data, as noted results



provided in Liu et al. (2013), who assimilated Air Force Weather Agency–National Aeronautics and Space Administration Snow Algorithm or (ANSA) SCE data for similar stations in the region. For example, the findings in Liu et al. (2013) for the best case indicate NSE improvement for Salcha, Little Chena and Chena at Fairbanks of 0.30, 0.31, and 0.06. Our study reports comparable NSE improvement values for some stations (Chatanika and

Goodpaster) for the months impacted by the adjustments, although the Salcha and Little Chena system differences are closer to those values reported for the raw MODIS data in Liu et al.'s (2013) study. The averaging approach and use of newly developed tools (ANSA, MODSCAG) applied in both studies appear to produce slightly superior results from that of MOD10A1. Further analysis is required to determine if cloud correction processes, such as those applied in the ANSA study, would act to reduce the impact of pixel shifting that is likely a major problem in Alaska

(Arsenault et al. 2014) and improve streamflow estimates further. Both studies indicate improved representation of internal snow pack and improvements in streamflow estimates for some basins, but not all, for these new iterations of the MODIS data.

Differences in the streamflow improvements provided by Liu et al. (2013) for the Salcha and Little Chena highlight some important variations between the two studies that should be considered. The first is that, as noted by the

authors, the model simulated streamflow estimates are biased and thus the improvements reported in the paper are still poor representations of the streamflow (Liu et al. 2013). The question then remains that if a model result without updated observations is already skillful, how much better or improved can the model be by added information (which carries its own uncertainty with it)? Perhaps the differences between the distributed model in Liu et al. (2013) versus the lumped models used in this study is also adding a buffer to the data improvements in the case of this study, and

limiting the amount of difference or improvement that MODIS SCE insertion can provide. Snow cover data appears to be improved at Interior locations within the model when compared to five different SNOTEL stations (Figure 5), particularly for the melt timing. However, the discharge values improved moderately given either MODIS input over the different periods analyzed, and in particular smaller changes are noted over the entire period of record (Table 4, Figure 8, 9). For the Chatanika basin, with limited observed data and poorer streamflow simulations however, the

improvements are closer to the values shown in the Liu study. These results suggest that skill can be added by introducing new observations when the models are performing poorly due to inadequate or low-quality records. Considering that there are numerous incomplete and low-quality gages throughout the high latitude regions of the globe, this result is of great value and indicates the utility of the MODIS SCE data in this regard.

Calibrations performed on the SACSMA model were limited in nature and targeted specifically at two parameters

exhibiting the most influence on improving discharge estimates during the melt season: MFMAX and TAELEV. These parameters control the air temperature and impact snow cover depletion by either increasing or retaining melt. Previously, the APRFC parameters were set to lower MFMAX values. The TAELEV parameter was not equal to the true elevation (ELEV) and set to different values for north and south aspects. For north-facing upper elevations, TAELEV was less than ELEV so temperatures were lapsed upward to simulate the slower melt rates and cooler

conditions. For south-facing aspects, TAELEV was set to greater than ELEV, so temperatures were lapsed downward to simulate increased melt from solar influence. Our updated parameterization using the MODIS data required an upward adjustment of these values because the areal depletion curve is no longer controlling the melt rate. Thus, SCE


present on northern, upper elevation slopes in the late spring must have higher melt rates applied to melt the snow with the correct timing. The primary reason that the areal depletion curves in SNOW17 differs from one that would be derived from actual measurements of SCE is that melt rates decline as SCE declines because the remaining snow is usually found in locations where snow melts at a slower rate, such as under canopies or on north facing slopes

(Anderson, 2006).

Adjustments to MFMAX across the north sub-basin units suggest that the modified areal depletion curves within SNOW17 underestimate snow covered area. At many of the sites, particularly when using the MODSCAG product, MFMAX for the northern sites had to be increased. This suggests that the APRFC run uses a lower value that attempts to account for cooler temperatures on the northern slopes by retaining the snow on these slopes for longer,

thus slowing runoff (Franz and Karsten, 2013). By more accurately representing conditions in the north sub-basin units of the watersheds, the MODIS runs required an increase in the snowmelt factor to allow for initiation of the melt on these slopes. MFMAX represents the dependency between the melt factor to account for a constant SCE curve used in the model; and the ability of the 'standard' SCE curves used in the APRFC SNOW17 to replicate the conditions of the melt properties within the basins (Shamir and Georgakakos 2007). As noted in Shamir and

Georgakakos (2007), there is considerable inter-annual variability in snow cover depletion and this variability is not represented when the standard APRFC model is applied. Therefore, by improving the internal physical processes in the model, the snowmelt timing should improve. However, this might not translate into improved discharge estimates because precipitation and temperature inputs could still be incorrect, and errors in forcing data that generate incorrect water equivalents for snow carry larger uncertainty bounds than that which can be addressed by changing the

weighting factors and timing of snowmelt by adjusting SCE, as undertaken in this study.

For the MOD10A1 calibration, fewer parameters were adjusted compared to the MODSCAG runs. The end result is that the MODSCAG data has improved streamflow simulations compared to the MOD10A1 result. The model parameters require greater adjustment for MODSCAG runs as a result of the variability between the two data sets compared to the APRFC baseline runs. As shown in Figure 4, the MODSCAG data have a different melt trajectory

for northern slopes and hold snow for longer on the south facing slopes of the Upper Chena River basin, while the MOD10A1 acts similarly to the APRFC melt trajectory for SWE data. This region is known to have variable melt timing based on south-facing slopes therefore the north and south slopes should be differentiated to reflect the physical processes occurring on the warmer south facing slopes compared to the cold, and often permafrost-dominated north facing slopes (Jones and Rinehart, 2010). Although MODSCAG improvement is noted for the

Chatanika and Goodpaster basins in the streamflow statistics, the results for both MODIS versions are overall very similar in this region (Figure 8). This may be due to the different canopy adjustments applied to the data sets, or because of the lack of a spectral end member for the boreal forest in MODSCAG (Painter et al. 2009). Regardless, it is not clear that one of these data sets is markedly improving streamflow estimates and it is possible that both approaches could be considerably useful as additional observations of SCE estimates for the region.

Two other means by which the CHPS framework can be altered to improve streamflow estimates are explored in this work. The interpolation over MODIS missing days can be altered easily in CHPS, however this had only a small effect on the streamflow results. The SCTOL, which allows for interaction between the model and the observed



MODIS SCE data, had an effect on streamflow and therefore may be a useful technique for the RFCs to apply during recalibration efforts to observed snow cover data. An advantage was noted between the MODSCAG with an SCTOL setting greater than to 0.25. However, the basins with the strongest improvement (Chatanika) over the APRFC run did not improve using an SCTOL greater than zero, which was because the baseline model performed so poorly

given the weakness of the underlying observed discharge data. Therefore, the RFCs may wish to selectively apply this parameter when basins have reliable observed information and the MODIS data can be utilized partially in conjunction with the model ADC and partially on the MODIS SCE observations.

## 5    Conclusions

Although complex tools and distributed models are available from the research community and in the CHPS system

to integrate observed snow cover extent data, the RFCs across the US are not, as of writing this paper, using these features in their operational river forecasting to estimate flooding and droughts. This study focuses on developing tools that can, with a minor amount of testing, be brought into the RFC's CHPS modeling framework and used to improve physical estimates of SCE across watersheds of interest. The method integrates information such as MODIS remotely sensed snow cover into the model framework using a simple calibration approach for the SNOW17 model,

and also provides some input regarding expected improvements and other possible parameters that may be introduced to improve forecasting and simulation of streamflow. Our recommendation it to incorporate MODIS data as an interim step, however, in the long run the RFCs should begin to use more complex models and data assimilation tools as the move towards the National Water Model proceeds.

In this work, we answer several outstanding questions regarding the application of MODIS data in the RFC models.

Basins with poor-quality streamflow observations benefited from the use of the MODIS SCE but improvements are also made to the internal snow timing estimates, observed in both the validation against SNOTEL data and also through the calibration that corrected the model parameters to better reflect the physical differences altering processes occurring on north and south facing slopes. Overall, minor differences were observed between MOD10A1 and MODSCAG data, however the MODSCAG data provided improvement over MOD10A1 when considering

average changes to streamflow simulations were observed in all basins. We observed limited impact of changing the interpolation length between missing days, although adjustments based on altering the interaction between the model and the observed MODIS SCE data did alter streamflow and therefore are useful to during recalibration efforts.

The utility of the MODIS data in CHPS goes beyond improvements to the streamflow; these tools can be used for a number of internal checks for SWE and SCE that are currently under way, such as the ingestion of data for ensemble

forecasts (NWS, 2012). This study opens the door for insertion of parameters via assimilation alongside developments such as physically-based model usage.

The observations of rapid change in the Arctic highlight important alterations to hydrological regimes that will be experienced in the subarctic Interior boreal forest of Alaska introduce a pressing need in Alaska to further understand the anticipated changes through modeling of major climate drivers of streamflow. The sparse observational network

in Alaska, along with the magnitude and rate of change necessitates the use of robust modeling tools to examine





these changes and their impacts on hydrology. However, due to the limited high- quality observations, and our lack of understanding of Arctic hydrologic processes (Woo et al. 2008, Prowse et al. 2016), process-based modeling approaches are limited in this environment. Therefore, we must apply available conceptual models with calibrations informed by observations, including remote sensing tools of SWE and SCE to examine these effects. In this way, we

5    will be able to define and quantify increasing impacts associated with these changes that lead to multi-scale risk to hydro-ecological systems, not only to the local and state resources, but also regionally and globally.

**6    Acknowledgements**

Bennett acknowledges support for this work from the Alaska Climate Science Center and the National Science and Engineering Research Council of Canada. Bennett, Balk, and Cherry acknowledge support from the GOES-R High

10   Latitude Proving Ground award NA08OAR432075. The authors thank the Alaska Pacific River Forecast Center staff for their collaborative efforts and the Arctic Region Supercomputer Center for the use of their supercomputer.



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





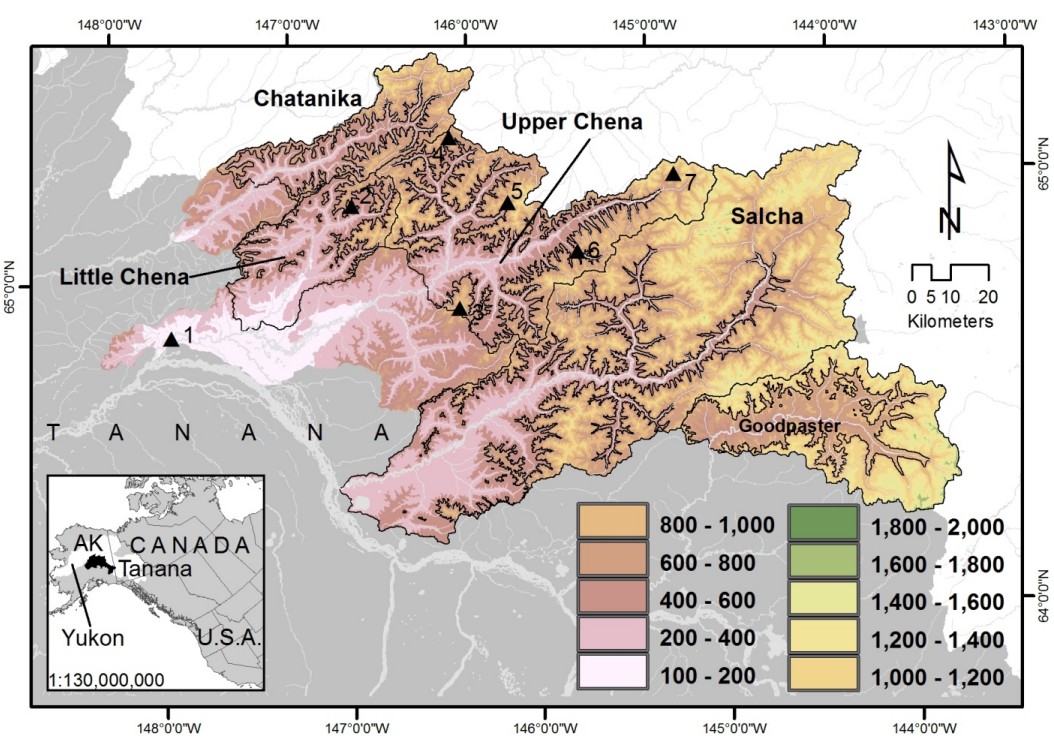

5   **Figure 1. Map of the five study basins with upper and lower divisions shown. Alaska**

6   **SNOTEL sites are shown with numbered black triangles: 1) Fairbanks International**

7   **Airport; 2) Little Chena Ridge; 3) Munson Ridge; 4) Mt. Ryan; 5) Monument Creek; 6)**

8   **Teuchet Creek; 7) Upper Chena (Table 2). Legend illustrates topographic variation**

9   **throughout the basins. Inset shows the Tanana River basin's location in the Yukon**

10  **watershed in proximity to Canada and the USA.**





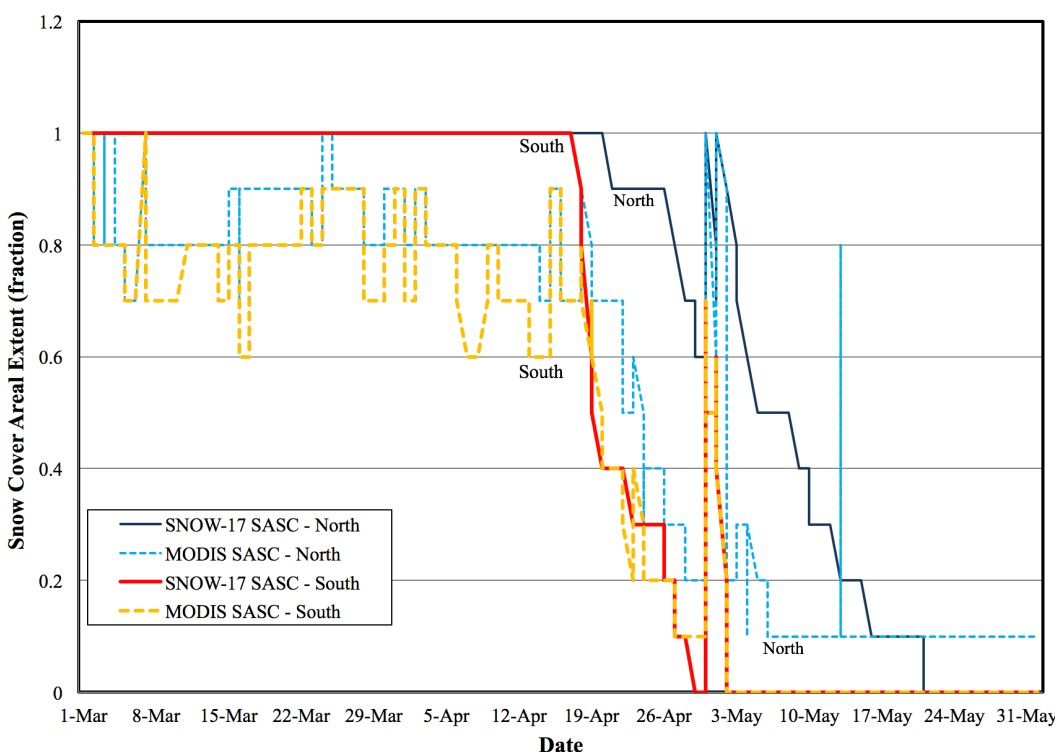

**Figure 2. Snow cover areal extent (SCE) for the Upper Chena river basin north slope from**

**SNOW17 and from MODIS. Large decreases in the MODIS SCE are observed compared**

**to the SNOW17 SCE.**





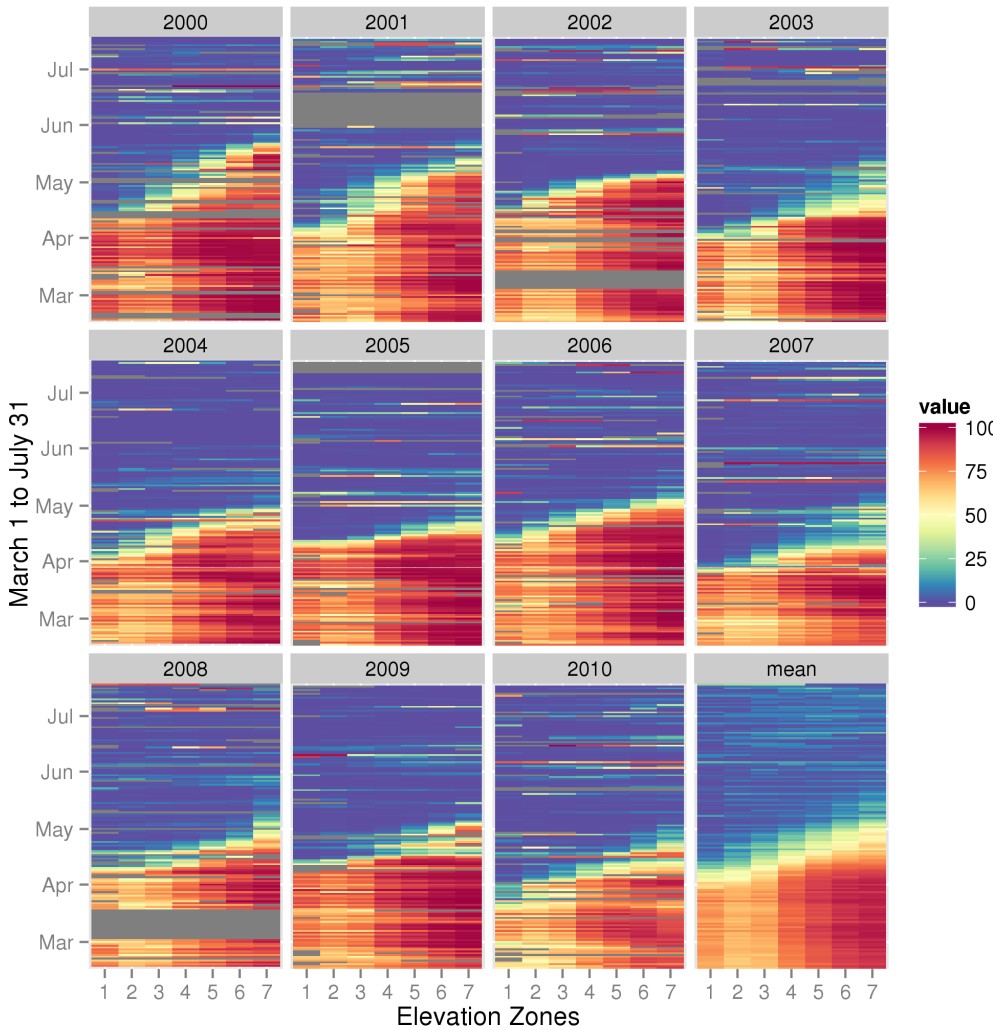

**Figure 3 Fractional snow cover extent based on MOD10A1 SCE average across all**

**watersheds divided into elevation zones. The years 2000 to 2010 are shown, with the mean**

**of all years in the final panel. Grey areas indicate dates when there is no SCE information**

**(i.e., cloud cover, missing sensor data).**





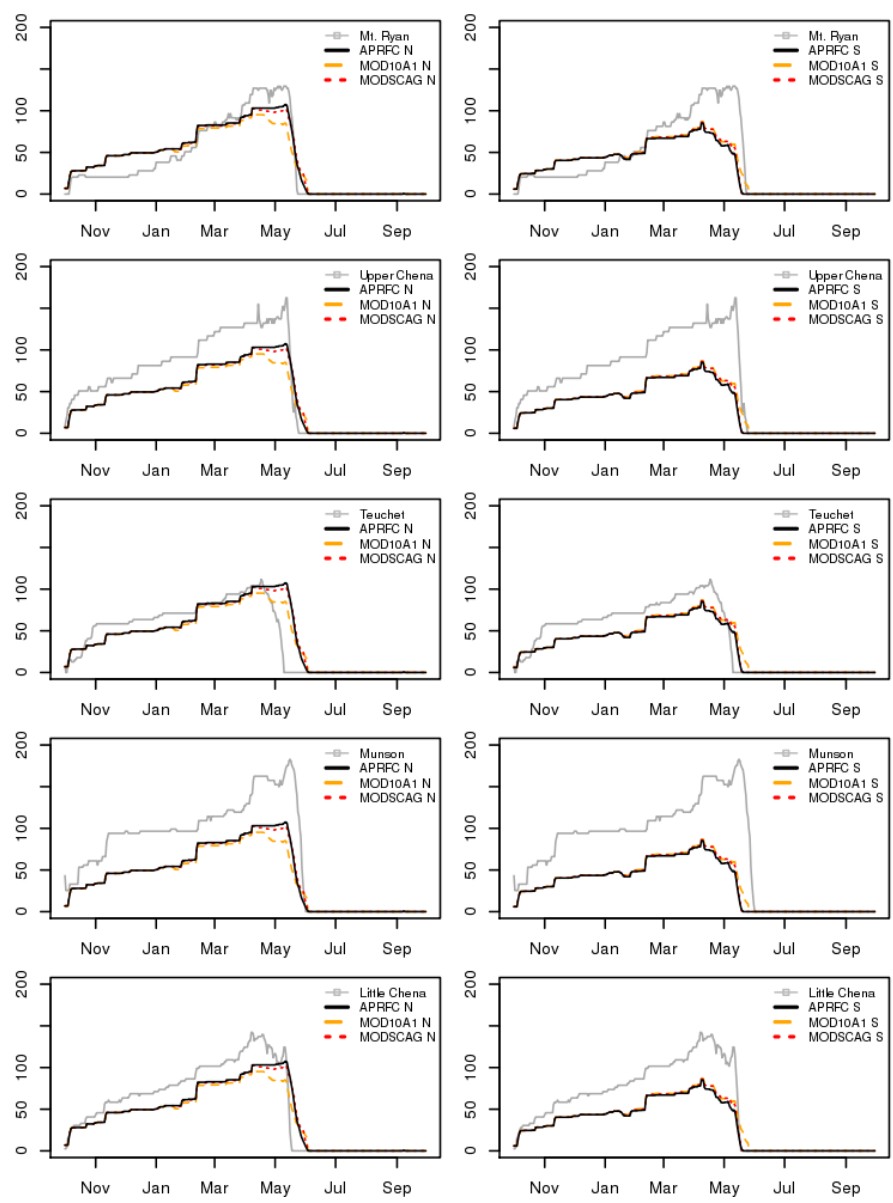

**Figure 4. Simulated SWE (mm) versus SNOTEL SWE (mm, grey line) for APRFC (solid**

**black line), MOD10A1 (orange dashed line), and MODSCAG (red dotted line) for October**

**1st, 2000 to June 30th, 2001. The Upper Chena River basin north slope is shown in the left**

**panels, and the south slope is shown in the right panels.**



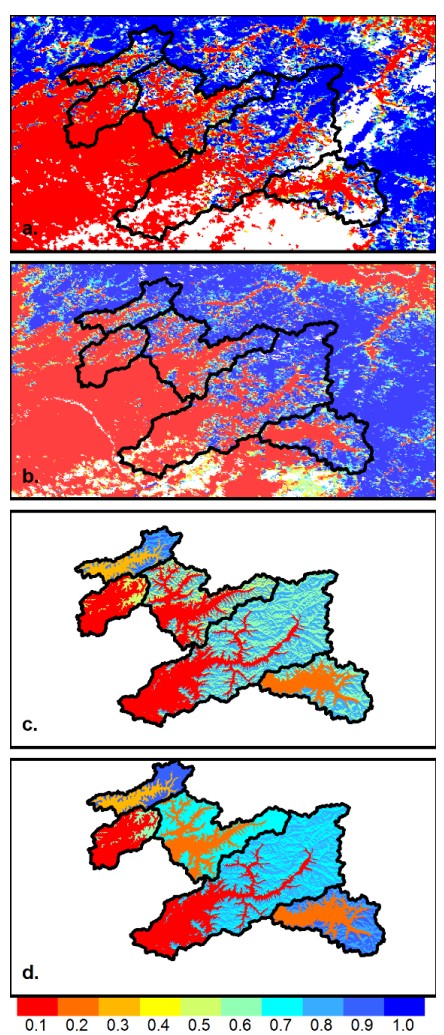

**Figure 5. Study area areal extent of snow cover in the CHPS model framework for: a)**

**MOD10A1 and b) MODSCAG, where white is either missing or cloud covered; and**

**elevation-averaged (lumped by elevation) snow cover extent based on: c) MOD10A1 and d)**

**MODSCAG. Values range from 0.1 to 1, or 10% to 100% snow cover. All panels show**

**results for May 15th, 2001.**

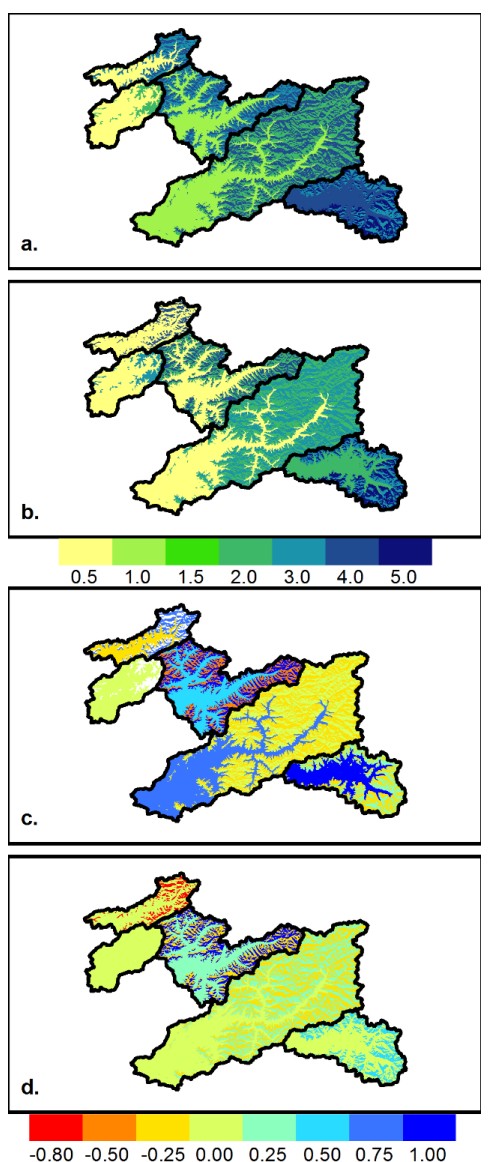

**Figure 6. Study area basin SWE (in) estimates in CHPS model framework for: a)**

**MOD10A1 and b) MODSCAG, and the difference between both SWE estimates and the**

**APRFC run (for positive values, MODIS is higher, for negative values, APRFC is higher;**

**Figures c) and d)). All panels show results for May 15th, 2001.**





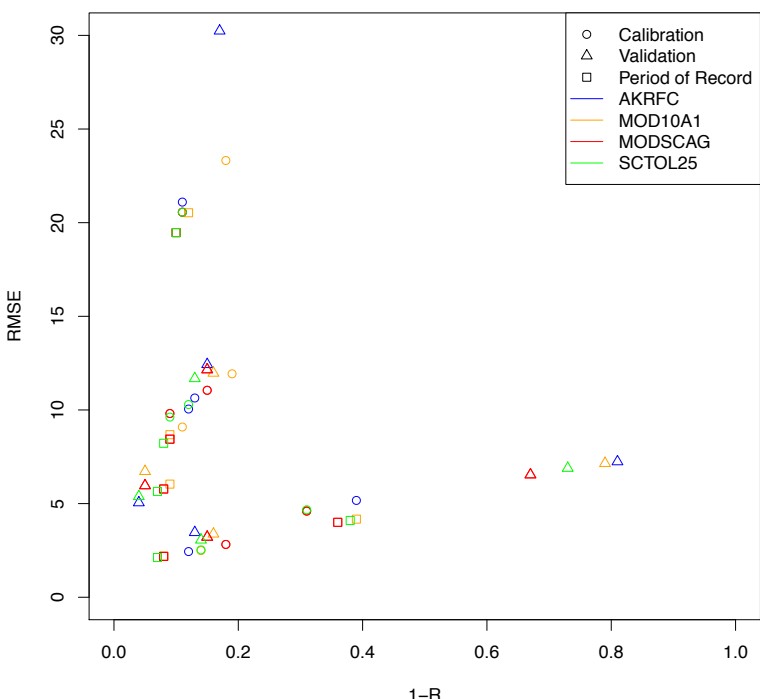

41

**Figure 7. RMSE plotted against 1-R for calibration (open circles), validation (open triangles) and period of record (open squares). Values are given for each of the five basins. Blue=APRFC, orange=MOD10A1, red=MODSCAG, and green=MODSCAG with SCTOL=0.25. Note that results cluster by basin.**

46





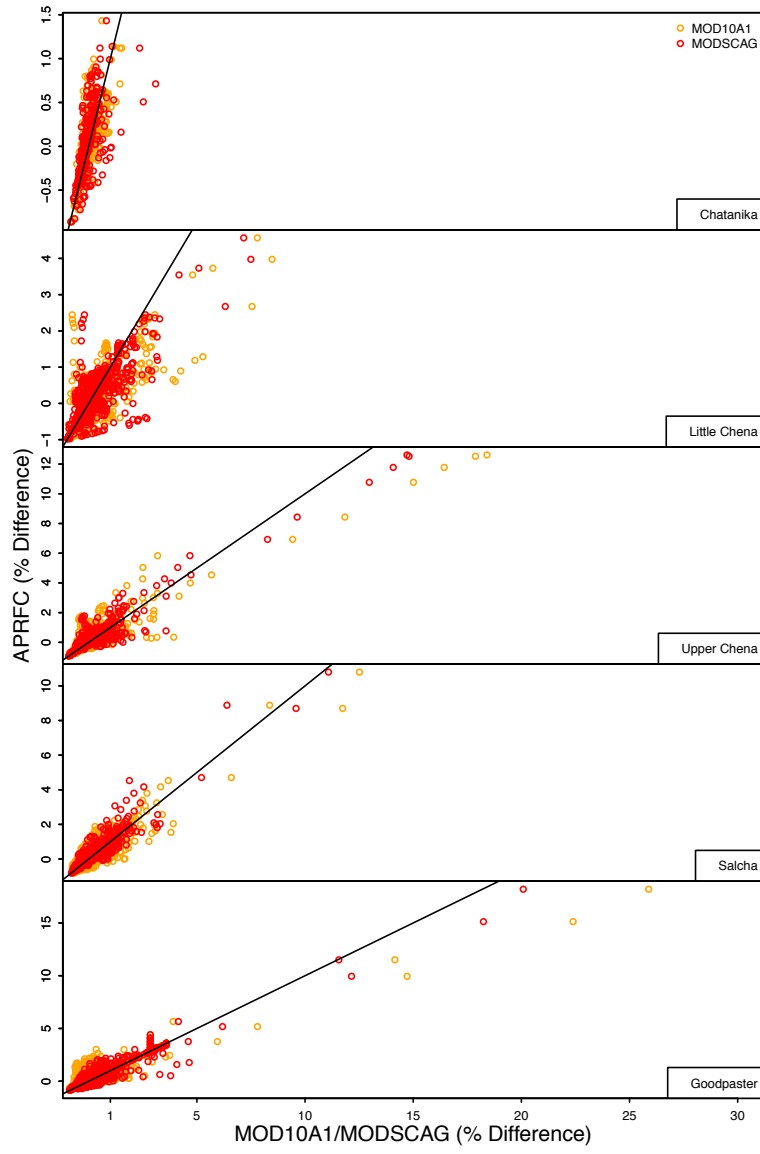

47

**Figure 8. Percent difference between observed streamflow and that modelled using**

**APRFC, MOD10A1 (orange), and MODSCAG (red). The APRFC percent difference (y-**

**axis) is plotted against the MOD10A1 and MODSCAG percent differences (x-axis). The 1:1**

**line is illustrated on plot for reference.**



52

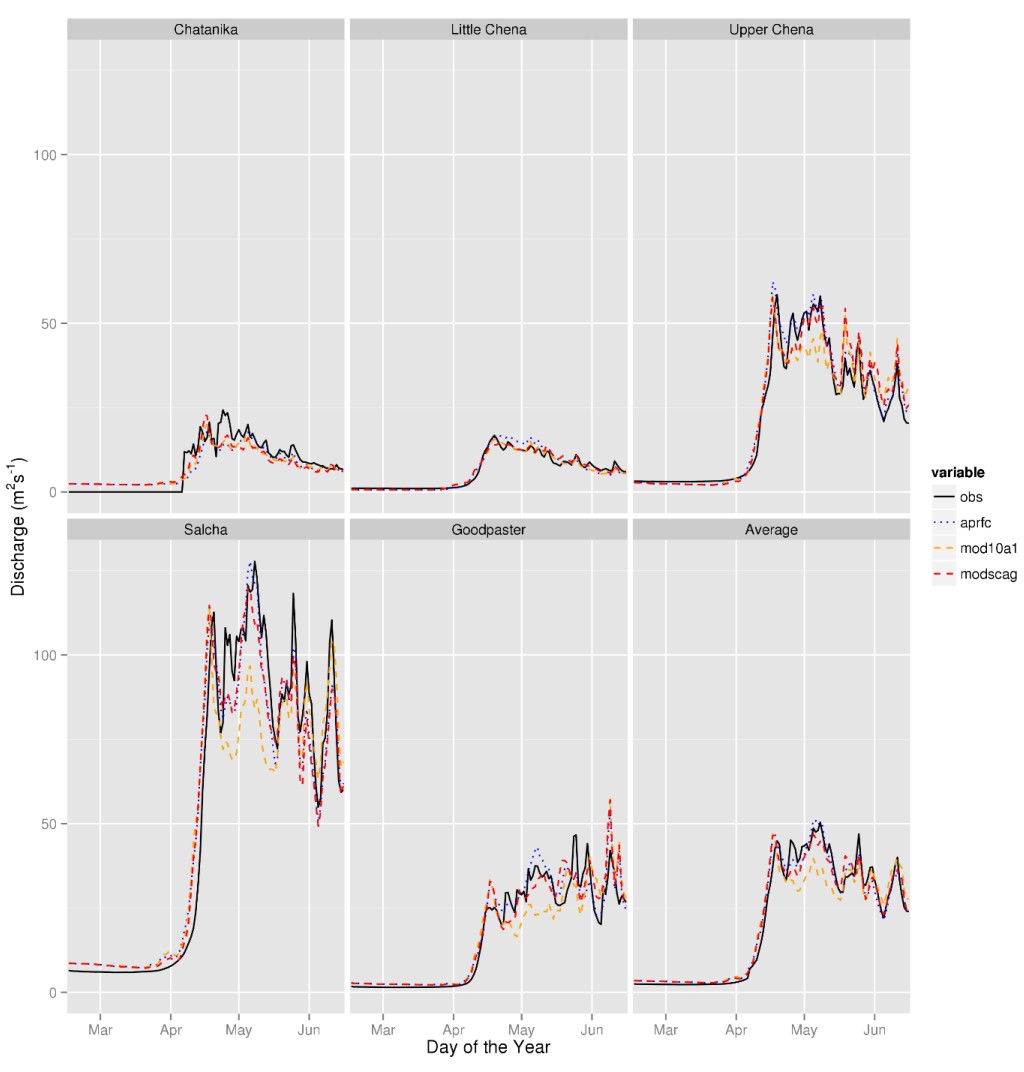

53

**Figure 9. Upper Chena River basin streamflow: observed (black line), simulated with**

**APRFC (blue dotted line), simulated with MOD10A1 (red dashed line), and simulated with**

**MODSCAG (yellow dashed line) for all years (2000-2010). The mean of all stations is**

**shown in the final panel.**



**Table 1.** Sub-basin characteristics, including name, sub-basin ID, area, elevation (mean and range below in brackets), T (average

monthly temperature; January and July), P (average annual total precipitation; seasonal total winter (October-March) and summer

(April-September) in brackets), Q (annual average daily discharge), % basin units (lower, N=north and S=south), % land cover (based

on majority cover values*). T, P, and Q calculated from the 2000-2010 water years.

| Name | Sub-basin ID | Sub-basin Unit | Area (km²) | Elevation (m) | T (ºC) | P (mm) | Q (m³/s/d) | Units N/S⁺ (%) | Land cover (%) |
|---|---|---|---|---|---|---|---|---|---|
| Chatanika at the Steese | CRSA2 | Lower | 395 | 475 (228 – 625) | -22.5 (12.6) | 96 (393) | 11 | 42 | 9 D, 83 C, 4S |
| | | Upper | 558 | 780 (548 – 1513) | -18.5 (11.9) | 116 (441) | | 25/33 | 0 D 76 C, 15 S / 2 D, 47 C, 39 S |
| Little Chena | CHLA2 | Lower | 802 | 380 (141 – 617) | -24.4 (13.7) | 97 (385) | 6 | 78 | 16 D, 78 C, 5 S |
| | | Upper | 225 | 721 (584 – 1230) | -21 (11.5) | 116 (464) | | 22 | 5 D, 72 C, 20 S |
| Upper Chena | UCHA2 | Lower | 973 | 466 (223 – 626) | -22.5 (12.7) | 75 (370) | 20 | 40 | 9 D, 84 C, 5 S |
| | | Upper | 1462 | 806 (553 – 1584) | -18.2 (11.6) | 103 (426) | | 29/31 | 2 D, 74 C, 17 S/ 10 D, 54 C, 33 S |
| Salcha | SALA2 | Lower | 1838 | 421 (194 – 624) | -23.9 (13.8) | 74 (364) | 44 | 32 | 18 D, 69 C, 10 S |
| | | Upper | 3900 | 924 (581 – 1768) | -19.3 (10.6) | 111 (475) | | 33/35 | 2 D, 63 C, 20 S / 7 D, 50 C, 31 S |
| Goodpaster | GBDA2 | Lower | 737 | 734 (411 – 967) | -20.6 (11.8) | 83 (389) | 14 | 42 | 2 D, 84 C, 12 S |
| | | Upper | 1036 | 1166 (873 – 1961) | -19.3 (10.1) | 104 (465) | | 29/29 | 5 C, 33 D 45 S / 2 D, 24 C, 56 S |

+Only upper units are divided into N and S units.

*D=deciduous, C=coniferous, S=shrubs



8    **Table 2.** SNOTEL stations, map identification, length of record, and observed average snow water equivalent (SWE) used for

9    validation of modeled SWE results. Average SWE is calculated as the average over the entire period of record.

| SNOTEL Station Name | Station code | Map ID | Record Length | Average SWE (mm) |
|---|---|---|---|---|
| Fairbanks F.O. | 47P03 | 1 | 1983-current | 446 |
| Little Chena Ridge | 46Q02 | 2 | 1981-current | 595 |
| Munson Ridge | 46P01 | 3 | 1980-current | 1016 |
| Mt. Ryan | 46Q01 | 4 | 1981-current | 639 |
| Monument Creek | 45Q02 | 5 | 1980-current | 554 |
| Teuchet Creek | 45P03 | 6 | 1981-current | 461 |
| Upper Chena | 44Q07 | 7 | 1985-current | 792 |





11   **Table 3.** SNOW17 model parameters. Sensitivity indicates whether a parameter has a major or minor influence on model output.

12   Minimum (Min) and maximum (Max) parameter values are provided.

| Parameter | Sensitivity | Description | Min | Max |
|---|---|---|---|---|
| SCF | Major | Snow correction factor that adjusts precipitation for gage deficiencies and processes not explicitly represented in the model (dimensionless) | 0.65 | 0.95 |
| MFMAX | Major | Maximum melt factor during non-rain periods occurring on June 21 (mm/°C/6 hrs) | 0.90 | 1.40 |
| MFMIN | Major | Minimum melt factor during non-rain periods occurring on December 21(mm/°C/6 hrs) | 0.20 | 0.20 |
| UADJ | Major | Average wind function during rain-on-snow periods (mm/mb) | 0.03 | 0.03 |
| SI | Major | Mean areal snow water equivalent below which there is less than 100% snow cover and the areal depletion curve is applied (mm) | 500 | 500 |
| NMF | Minor | Determines the amount of energy exchange that occurs when melt is not taking place at the snow surface. Maximum negative melt factor (mm/°C/6 hrs). | 0.15 | 0.30 |
| DAYGM | Minor | Constant melt rate at the snow/soil interface (mm) | 0.00 | 0.00 |
| MBASE | Minor | Base air temperature for non-rain melt computations (°C) | 0.00 | 00 |
| PXTEMP | Minor | Air temperature threshold at which precipitation is defined as rain or snow (°C) | 1.70 | 1.70 |
| PLWHC | Minor | Maximum liquid water holding capacity of the snowpack (decimal fraction) | 0.05 | 0.05 |
| TIPM | Minor | Antecedent temperature index (dimensionless) | 0.10 | 0.10 |
| PXADJ | Minor | Adjustment factor for precipitation, must be between 0.0 and 1.0 (dimensionless) | 0.97 | 1.21 |
| TAELEV | Minor | Elevation at which the air temperature time series is collected (m) | 380 | 1267 |
| ELEV | Minor | Average sub-basin elevation (m) | 380 | 1167 |
| SCTOL | Minor | Tolerance used when updating water equivalent or areal extent of snow cover with observed data. Range is 0.0 to 1.0. Updates when \|Simulated-Observed\| > Tolerance*Observed (dimensionless) | 0.00 | 0.05 |





15   **Table 4.** April-May-June monthly calibration (Cal), validation (Val) and the period of record

16   (Per., 1999-2010) statistics (MAE=mean absolute error (m³/sec), NSE=Nash Sutcliffe efficiency

17   (unitless), PBIAS=flow bias (%), R=correlation coefficient (unitless), and RMSE=root mean

18   squared error (m³/sec) for APRFC, MOD10A1, and MODSCAG modeled discharge for all

19   basins.

| | | APRFC | | | MOD10A1 | | | MODSCAG | | |
|---|---|---|---|---|---|---|---|---|---|---|
| | **Stat** | **Cal** | **Val** | **Per** | **Cal** | **Val** | **Per** | **Cal** | **Val** | **Per** |
| *CRSA2* | **MAE** | 3.96 | 4.73 | 3.07 | 3.39 | 4.66 | 2.96 | 3.37 | 4.22 | 2.87 |
| | **NSE** | 0.10 | -0.87 | -0.04 | 0.28 | -0.82 | 0.03 | 0.29 | -0.53 | 0.11 |
| | **PBias** | -17.28 | -25.48 | -13.08 | -16.37 | -26.83 | -13.07 | -16.13 | -25.71 | -13.27 |
| | **R** | 0.61 | 0.19 | 0.58 | 0.69 | 0.21 | 0.61 | 0.69 | 0.33 | 0.64 |
| | **RMSE** | 5.17 | 7.24 | 4.31 | 4.62 | 7.15 | 4.17 | 4.60 | 6.54 | 4.00 |
| | | | | | | | | | | |
| *CHLA2* | **MAE** | 1.85 | 2.88 | 1.57 | 2.00 | 2.84 | 1.59 | 2.09 | 2.47 | 1.52 |
| | **NSE** | 0.74 | 0.58 | 0.81 | 0.73 | 0.60 | 0.81 | 0.66 | 0.64 | 0.82 |
| | **PBias** | 4.29 | 4.84 | -2.32 | -4.14 | -0.65 | -5.06 | 0.56 | 3.12 | -2.84 |
| | **R** | 0.88 | 0.87 | 0.93 | 0.86 | 0.84 | 0.92 | 0.82 | 0.85 | 0.92 |
| | **RMSE** | 2.44 | 3.46 | 2.20 | 2.49 | 3.38 | 2.20 | 2.82 | 3.21 | 2.18 |
| | | | | | | | | | | |
| *UCHA2* | **MAE** | 9.12 | 8.22 | 5.34 | 9.15 | 8.01 | 5.40 | 8.75 | 8.82 | 5.37 |
| | **NSE** | 0.71 | 0.62 | 0.81 | 0.63 | 0.65 | 0.80 | 0.69 | 0.64 | 0.81 |
| | **PBias** | 16.76 | 0.39 | 0.21 | 10.46 | -4.59 | -1.05 | 14.42 | -0.48 | -0.10 |
| | **R** | 0.87 | 0.85 | 0.91 | 0.81 | 0.84 | 0.91 | 0.85 | 0.85 | 0.91 |
| | **RMSE** | 10.64 | 12.43 | 8.43 | 11.93 | 11.97 | 8.68 | 11.05 | 12.15 | 8.44 |
| | | | | | | | | | | |
| *SALA2* | **MAE** | 17.66 | 21.93 | 12.31 | 19.2 | 24.81 | 12.94 | 17.25 | 23.4 | 12.45 |
| | **NSE** | 0.69 | 0.63 | 0.80 | 0.63 | 0.53 | 0.78 | 0.71 | 0.60 | 0.80 |
| | **PBias** | 17.21 | -14.98 | 0.35 | 9.85 | -19.07 | -1.28 | 15.18 | -15.77 | -0.27 |
| | **R** | 0.89 | 0.83 | 0.90 | 0.82 | 0.78 | 0.88 | 0.89 | 0.81 | 0.90 |
| | **RMSE** | 21.10 | 30.24 | 19.27 | 23.32 | 34.20 | 20.53 | 20.56 | 31.57 | 19.47 |
| | | | | | | | | | | |
| *GBDA2* | **MAE** | 7.00 | 3.91 | 3.62 | 6.57 | 5.28 | 3.93 | 6.45 | 4.21 | 3.63 |
| | **NSE** | 0.45 | 0.90 | 0.84 | 0.55 | 0.83 | 0.82 | 0.47 | 0.86 | 0.83 |
| | **PBias** | 28.10 | -11.17 | 1.46 | 14.41 | -17.60 | -1.56 | 25.89 | -12.19 | 0.83 |
| | **R** | 0.88 | 0.96 | 0.92 | 0.89 | 0.95 | 0.91 | 0.91 | 0.95 | 0.92 |
| | **RMSE** | 10.05 | 5.05 | 5.66 | 9.09 | 6.72 | 6.04 | 9.81 | 5.96 | 5.78 |



20   **Table 5.** SNOW17 parameters for the MOD10A1 calibration. North (N), south (S) and lower (L) sub-basins are described. For each

21   sub-basin, the first column indicates the parameter value in the APRFC calibration and the second column indicates the parameter

22   value used in the MODIS calibration. Bolded values indicate where the MODIS value differs from the APRFC value.

| Parameter | Sensitivity | N | | S | | L | | N | | S | | L | | U | | L | |
|---|---|---|---|---|---|---|---|---|---|---|---|---|---|---|---|---|---|
| | | | CRSA2 | | | | | | UCHA2 | | | | | | CHLA2 | | |
| MFMAX | Major | 1.00 | 1.00 | 1.40 | 1.40 | 1.00 | **1.40** | 0.90 | 0.90 | 1.40 | 1.40 | 1.00 | 1.00 | 0.90 | **1.40** | 1.30 | **1.40** |
| NMF | Minor | 0.30 | **0.15** | 0.30 | **0.20** | 0.30 | **0.20** | 0.30 | **0.15** | 0.30 | **0.20** | 0.30 | **0.20** | 0.20 | 0.20 | 0.20 | 0.20 |
| TAELEV | Minor | 665 | **865** | 1088 | **988** | 474 | 474 | 708 | **908** | 1002 | **908** | 465 | 465 | 720 | **820** | 380 | **480** |
| SCTOL | Minor | 0.05 | **0.00** | 0.05 | **0.00** | 0.05 | **0.00** | 0.05 | **0.00** | 0.05 | **0.00** | 0.05 | **0.00** | 0.05 | **0.00** | 0.05 | **0.00** |
| | | | SALA2 | | | | | | GBDA2 | | | | | | | | |
| MFMAX | Major | 0.90 | 0.90 | 1.40 | 1.40 | 1.00 | 1.00 | 0.90 | **1.00** | 1.40 | 1.40 | 1.00 | **0.90** | | | | |
| NMF | Minor | 0.30 | **0.15** | 0.30 | **0.20** | 0.30 | **0.20** | 0.30 | **0.15** | 0.30 | **0.20** | 0.30 | **0.20** | | | | |
| TAELEV | Minor | 823 | **1023** | 1123 | 1123 | 420 | 420 | 863 | **1167** | 1267 | 1267 | 734 | 734 | | | | |
| SCTOL | Minor | 0.05 | **0.00** | 0.05 | **0.00** | 0.05 | **0.00** | 0.05 | **0.00** | 0.05 | **0.00** | 0.05 | **0.00** | | | | |



24  **Table 6.** SNOW17 parameters for the MODSCAG calibration. North (N), south (S) and lower (L) sub-basins are described. For each

25  sub-basin, the first column indicates the parameter value in the APRFC calibration and the second column indicates the parameter

26  value used in the MODIS calibration. Bolded values indicate where the MODIS value differs from the APRFC value.

| Parameters | Sensitivity | N | | S | | L | | N | | S | | L | | U | | L | |
|---|---|---|---|---|---|---|---|---|---|---|---|---|---|---|---|---|---|
| | | CRSA2 | | | | | | UCHA2 | | | | | | CHLA2 | | | |
| MFMAX | Major | 1.00 | **1.20** | 1.40 | 1.40 | 1.00 | **1.20** | 0.90 | **1.00** | 1.40 | 1.40 | 1.00 | **0.90** | 0.90 | 0.90 | 1.30 | **1.20** |
| NMF | Minor | 0.30 | **0.15** | 0.30 | **0.20** | 0.30 | **0.20** | 0.30 | **0.15** | 0.30 | **0.20** | 0.30 | **0.20** | 0.30 | **0.20** | 0.30 | **0.20** |
| TAELEV | Minor | 665 | 665 | 1088 | 1088 | 474 | 474 | 708 | **702** | 1002 | **902** | 465 | 465 | 720 | 720 | 380 | **580** |
| SCTOL | Minor | 0.05 | **0.00** | 0.05 | **0.00** | 0.05 | **0.00** | 0.05 | **0.00** | 0.05 | **0.00** | 0.05 | **0.00** | 0.05 | **0.00** | 0.05 | **0.00** |
| | | SALA2 | | | | | | GBDA2 | | | | | | | | | |
| MFMAX | Major | 0.90 | **1.00** | 1.40 | 1.40 | 1.00 | 11 | 0.90 | **1.00** | 1.40 | 1.40 | 1.00 | **0.90** | | | | |
| NMF | Minor | 0.30 | **0.15** | 0.30 | **0.20** | 0.30 | **0.20** | 0.30 | **0.15** | 0.30 | **0.20** | 0.30 | **0.20** | | | | |
| TAELEV | Minor | 823 | **923** | 1123 | **1023** | 420 | 420 | 863 | **863** | 1267 | **1163** | 734 | 734 | | | | |
| SCTOL | Minor | 0.05 | **0.00** | 0.05 | **0.00** | 0.05 | **0.00** | 0.05 | **0.00** | 0.05 | **0.00** | 0.05 | **0.00** | | | | |





29    **Table 7.** Comparison between RMSE (%) and NSE (in brackets) for April-May-June using

30    SCTOL values of 0.25, 0.50 and 0.75. Absolute differences are calculated from the MODSCAG

31    base run.

| SCTOL | | CRSA2 | UCHA2 | CHLA2 | SALA2 | GBDA2 |
|---|---|---|---|---|---|---|
| **0.25** | **Cal.** | -2 | 18 | 12 | 19 | -2 |
| | | (-0.03) | (0.13) | (0.08) | (0.16) | (-0.03) |
| **0.50** | | -10 | -9 | -1 | 8 | 12 |
| | | (-0.29) | (-0.05) | (-0.01) | (0.07) | (0.03) |
| **0.75** | | -4 | 4 | 2 | 6 | 5 |
| | | (-0.08) | (0.02) | (0.01) | (0.03) | (0.01) |
| **0.25** | **Val.** | -11 | 19 | 15 | 18 | -6 |
| | | (-0.17) | (0.14) | (0.1) | (0.15) | (-0.06) |
| **0.50** | | -15 | -14 | -2 | 6 | 12 |
| | | (-0.45) | (-0.08) | (-0.02) | (0.06) | (0.03) |
| **0.75** | | -8 | 3 | 2 | 6 | 4 |
| | | (-0.15) | (0.01) | (0.01) | (0.03) | (0.01) |
| **0.25** | **Per.** | -10 | 21 | 12 | 19 | -7 |
| | | (-0.15) | (0.15) | (0.08) | (0.16) | (-0.08) |
| **0.50** | | -11 | -20 | -12 | 7 | 17 |
| | | (-0.34) | (-0.12) | (-0.09) | (0.06) | (0.04) |
| **0.75** | | -7 | 1 | -1 | 6 | 4 |
| | | (-0.13) | (0.01) | (0) | (0.03) | (0.01) |

