# Peer review of "Using MODIS estimates of fractional snow cover area to improve streamflow forecasts in Interior Alaska"

_Hydrology and Earth System Sciences, 2018_

## Referee Comment (RC1) · E.H. Bair (Referee) · 16 May 2018

Using MODIS estimates of fractional snow-covered area to improve streamflow forecasts in Interior Alaska by Bennett et al. examines improvements in model skill when remotely-sensed snow-covered area estimates are used to model streamflow, compared to model runs where model-generated areal depletion curves are used. For this study, two MODIS-derived snow-covered area products were used, MOD10A1 and MODSCAG.

This study is a nice assessment of the use of remotely-sensed snow cover products with the new CHPS modeling framework for several watersheds in the interior of Alaska. This study demonstrates the improvements as well as pitfalls of using areal depletion curves vs. remotely sensed snow-covered area. The authors find that using remotely sensed snow-covered area yields modest improvements in some basins, especially the sparsely measured ones, but not in others. These findings agree with previous studies, which the authors cite. Overall, the techniques are well researched and the findings are sound, but I have a few major concerns that I would like to see addressed prior to publication:

1) In most of the cited publications, e.g. Painter et al. (2009); Rittger et al. (2013), what is referred to in this manuscript as snow covered extent is called fractional snow-covered area, or fSCA. Since MODSCAG and MOD10A1 are both fractional products, fractional snow-covered area is a more accurate term than snow-covered extent. Thus, I suggest changing snow-covered extent to fractional snow-covered area to align with most other publications.

2) What is really needed for model input is the total volume of snow water equivalent (SWE). The fSCA contains no information on depth. Among other problems, as the authors point out, when fSCA reaches 100%, it gives little information about the snow volume. I realize that there is no good direct SWE estimate for model input, however there have been many attempts to create basin-wide SWE estimates, for example by fusing snow telemetry estimates with fSCA (Fassnacht et al., 2003; Dozier et al., 2016; Schneider and Molotch, 2016). It would be worthwhile to at least discuss why fSCA only was chosen to improve the streamflow forecasts.

3) The interpolation, filtering, and smoothing of both MOD10A1 and MODSCAG is barely mentioned in the text and the supplement. Snow-cloud discrimination and how MODIS data are smoothed is a critical step that the authors have, at the least, not fully addressed. Likewise, viewing geometry also greatly affects the accuracy of MODIS surface reflectance (Tan et al., 2006), which both MOD10A1 and MODSCAG are based on. I recommend the following two studies as examples of different smoothing approaches for snow cover from MODIS, Dozier et al. (2008); Morriss et al. (2016). I would like to know how the authors' approach compares to these two smoothing techniques.

I have included minor comments as an annotated PDF. Several citations in the text were not in the bibliography. Thus, I suggest the authors carefully check that the citations in the text correspond to those in the bibliography. If the authors have any questions about my review, I encourage them to contact me at nbair@eri.ucsb.edu.

Sincerely,

Ned Bair

Earth Research Institute
University of California, Santa Barbara

[revised manuscript text omitted]

---

## Referee Comment (RC2) · Anonymous Referee #2 · 11 Jun 2018

In their study "Using MODIS estimates of fractional snow cover extent to improve streamflow forecasts in Interior Alaska" Bennett et al. investigate the value of two MODIS-derived snow cover area products (MOD10A1 and MODSCAG) to improve streamflow simulations in the interior of Alaska as compared to simulations where model-generated areal depletion curves are used. The authors conclude that there is only marginal improvement when evaluating the model performance with metrics such as NSE, RMSE etc., but argue that the MODIS derived snow covered area products might be valuable particularly in regions with sparse or poor quality observations. The methods and findings are sound and the article is generally written in a clear, concise and very structured way.

Nevertheless, I have som minor comments, which I would like to see addressed prior to publication:

General comments

- The basins with the sparsest streamflow observation had the greatest improvement in streamflow simulations. (P1L13,P14L 25-26) –> could indeed the sparse data be the main reason for the improvement rather than the product? This could be tested with a little model experiment adding some data gaps and see how the performance measure is sensitive to that maybe for the basin with the longest observations

- Why do the authors include the Chatanika catchment, when it shows very poor performance both using the model generated and the MODIS product derived snow cover extend? Please clarify.

- Move more detailed description of the derivation and differences between the MODIs products (interpolation, filtering, and smoothing) from the supplements in the main study.

Minor/technical comments

- There are passages in the article where fill words like utmost, great, very are used abundantly. In my opinion, it would help to go through the article and check where these are really needed and where they could be dropped.

- There is a mixed use of watershed and basin. Do the authors use it with equivalent meaning? If so, then use only one term throughout the paper, if not please clarify why the two terms are used.

- Often model runs could be replaced for better clarity with SWE simulations (e.g. P10L30) or streamflow simulations, respectively.

[Figure]

- Please make the units consistent (sometimes there is sec, sometimes there is s, etc.)

P1L29/30 did both extent and duration decrease by the same percentage?

P1L35 Delete this sentence

P2L3 Extremes –> Extreme

P3L14 in which P3L22/23 what does it mean that they perform better, better than in other regions or these models are better in these regions than the other models. Please clarify.

P3L33 Date missing in reference

P4L33 delete above the Steese Highway ( I do not see the relevance of this information)

P5L23 delete at the Steese Highway site

P5L35 add eq. 1 in brackets

P6L1 does still really need to be expressed in feet?

P6L17 how sensitive are streamflow simulations to this lapse rate? What motivates the assumption that the fixed lapse rate of 0.6°C/100m holds?

P6L30 delete "and is set to . . . "already mentioned above

P6L33 mm/mb/6hr is that the unit for rain on snow, then move to melt from rain of snow earlier in the sentence

P7L12-15 It might be helpful to see a sketch of how the ADC works

P7L16 add a reference to this look up table

P7L29 change "produces streamflow simulates" to "simulates streamflow"

P8L12 delete study

P8L30 to P910 all of these statistics are well known, I think it would be sufficient to just add some reference for each, else, I would summarize them in a separate table. In case the authors want to keep the equations, check the equations carefully: MAE has to additionally be divided by the number of observations, Spearman correlation coefficient could simply be written in a simpler form.

P9L10 make units consistent

P10L1 delete "for reasons that are discussed in the following section"

P10L5 why it is the maximum recommended value and who recommended it? Maybe refer again to the table with parameter ranges

P10L14 What does a more rigorous calibration mean here?

P11L10 Why for May 15th 2001?

P12L6 1-R

Figures

- Most figures are difficult to read when printed in black and white. This could be improved easily by adapting the color palette.

- Figure1: not all elevation classes are used in the map. Units are missing for the elevation zones. Drop last sentence in caption.

- Figure2: what happens to MODIS SASC North between May 10 and 17?

- Figure3: It seems there is a difference in the fractional snow cover extend seasonal development for the years that are used for calibration and the years that are used for validation. Is this also the case for each individual catchment, or is one catchment causing this difference? Is it ok to shift the year in validation, calibration period in the Chatanika catchment compared to the other catchments?

- Figure9: could the time be extended spanning April to September as also used in Table 1? Would it be possible to add quantiles of the streamflow to get an idea about the range per season? Unit should be m3/s, I do not understand what the average of all basins can tell me.

  Tables

  – Table1: units: elevation m a.s.l.; Q m3/s?

- Table2: maybe replace current with the last year included
- Table3: SCF Max values seem messed up
- Table4: mention that period of record is not the same for each catchment!
* * *

---

## Referee Comment (RC3) · S. Déry (Referee) · 12 Jun 2018

Review of "Using MODIS estimates of fractional snow cover extent to improve streamflow forecasts in Interior Alaska" by K. E. Bennett, J. E. Cherry, B. Balk and S. Lindsey

Submitted to *Hydrology and Earth System Sciences*

Manuscript Number: HESS-2018-96

**Summary:**

In this paper, the authors employ daily Moderate Resolution Spectroradiometer (MODIS) fractional snow cover extent (SCE) data to improve streamflow simulations in several Alaskan sub-watersheds of the Tanana River. The study period covers 2000-2010 with simulations with the SAC-SMA conceptual rainfall-runoff model that also incorporates the one-layer SNOW17 model for the representation of snowpack conditions. Runoff simulations that include MODIS-derived snow areal depletion curves (ADCs) in SNOW17 are compared with baseline simulations with the standard model formulation for ADCs in the five sub-basins of the Tanana River. The authors conclude that the assimilation of the MODIS SCE data leads to better representation of snow conditions and runoff simulations in Interior Alaska.

This paper presents interesting results on the potential application of MODIS SCE data in operational models for improved runoff simulations in Interior Alaskan watersheds where in situ data remain sparse. The paper is generally well-written and illustrated, but the paper requires some revisions prior to publication. The following provides a list of suggestions that may be helpful to the authors in revising their paper:

**General Comments:**

1) The paper includes non-metric units including feet for elevations and inches for snow water equivalent (SWE). Please convert all non-metric units to metric and adjust Equation (1) accordingly.

2) A considerable amount of effort has been placed into ingesting the MODIS SCE data into SAC-SMA model simulations of runoff in five sub-watersheds of the Tanana River. The authors should be commended for this effort. Nonetheless, the results shown in Figure 9 show little differences between the simulations that incorporate the MODIS data versus those with the standard model formulation. Table 4 confirms there are only very modest gains to be made by ingesting the MODIS data into the runoff simulations. As stated by the authors, more significant gains would be obtained by having more accurate forcing data (air temperature and precipitation) in the remote and complex terrain of these Alaskan watersheds. Further to this, SNOW17 incorporates only one snow layer which may miss some of the snow dynamics at play within the thin snowpack layer than interacts with the atmosphere. As such, why spend so much effort in trying to improve the runoff simulations with the data assimilation strategy when more

significant gains may be obtained by improving other aspects of the modeling framework?

3) Why does the study period cover only 2000-2010 when MODIS data are available up to present? Further to this, how are gaps in the MODIS data in-filled? For instance, persistent cloud cover can lead to a significant reduction in the available snowcover data from optical remote sensing. Is any gap-filling procedure used to address this issue (see for example Hall et al., 2010 and Tong et al., 2009).

4) Hydrological simulations such as those presented in Figure 9 are averaged over 10 water years. Results for each individual year should also be presented to illustrate the model's ability to represent interannual variability in the discharge patterns.

5) The references need to be fully revised and presented in the journal's standard format.

6) Note that Déry et al. (2005) used MODIS ADCs to improve their simulations of runoff on the Alaskan North Slope and may be a relevant reference to this study.

**Specific Comments:**

1) P. 2, Abstract: Include the study period within the abstract.

2) P. 2, lines 5 and 25: Define "US".

3) P. 2, lines 29/30: Have both snow cover extent and duration in Alaska indeed declined by 18% from 1966 to 2012?

4) P. 2, line 31: What aspect of permafrost has declined in response to warmer air temperatures in Alaska? Its depth, extent, or other characteristic?

5) P. 2, line 34: Change to "North American".

6) P. 3, line 3: "Extremes" should be singular.

7) P. 3, line 16: Delete the extra "model output".

8) P. 3, line 20: Define "NOAA".

9) P. 3, line 30: Change to "these data have".

10) P. 5, line 15: Why does the study period end in 2010 although MODIS data are available up to present?

11) P. 5, line 27: Define "SWE" upon first usage rather than in the following line.

12) P. 5, line 35: Perhaps number the equations, depending on the journal's formatting guidelines. Convert the equation to metric units and ensure the elevation $e$ is in meters, not feet.

13) P. 6, line 3: Define "SAC-SMA".

14) P. 6, line 17: Should the air temperature lapse rate be 0.6°C/100 m? Insert a space in "100 m".

15) P. 6, line 23: The journal may prefer dates in a format such as "21 December".

16) P. 6, line 29: Insert a space in "100 m".

17) P. 6, line 31: What atmospheric temperature is used to compute incoming longwave radiation with the Stefan-Boltzmann Law?

18) P. 6, line 32: Why assume a constant relative humidity (RH) at 90%? Is this relative to a water (and not an ice) surface even when air temperatures are subfreezing? How does RH enter the calculation of the simplified energy balance, through the latent heat flux?

19) P. 6, line 33: How can wind have units of "mm/mb/6 hr"?
20) P. 6, lines 34/35: Write "snowpack" as one word.
21) P. 8, line 30: Revise to: "Three additional objectives"
22) P. 9, lines 1 through 9: Equations numbers run on two lines and are missing for the last three equations.
23) P. 9, line 10: The units should be "$m^3/s$".
24) P. 9, line 17: Provide probability values for all correlation coefficients reported in the study.
25) P. 10, line 18: What are the units for snow density, listed here only as 0.2?
26) P. 10, line 19: Insert a space in "6 hr".
27) P. 10, line 35: Insert a space in "850 m".
28) P. 10, line 36: Should this be "SNOW17's"?
29) P. 11, lines 1 and 11: Write "snowpack" as one word.
30) P. 11, line 10: Date format may need to be revised to "15 May 2001". Please also change to "is shown in Figure 5b".
31) P. 11, line 13: Change to "watershed's".
32) P. 11, lines 20 to 22: Convert SWE from inches to mm.
33) P. 11, line 33: Change to "improve".
34) P. 12, lines 4/5 and 13/15: Avoid sentences that just describe the figures – this is what figure captions are for.
35) P. 12, line 35: Delete "Because this."
36) P. 13, line 14: Change to "SNOW17's".
37) P. 13, line 17: Revise to "data are temporally".
38) P. 14, line 11: Write "snowpack" as one word.
39) P. 14, line 19: Change to "are adding".
40) P. 14, line 20: Change to "data appear".
41) P. 15, line 22: Change to "have improved".
42) P. 16, line 11: For consistent language, change to "floods and droughts".
43) P. 16, line 27: Delete "to" before "during".
44) P. 16, lines 32 to 34: This sentence is long and confusing. Consider revising it and perhaps dividing it into two sentences.
45) P. 17, line 1: Delete the space after the hyphen in "high-quality".
46) P. 17, line 8: Change to "Natural Sciences".
47) P. 18, line 1: Note that the references do not generally follow the format used by HESS; for instance, journal names should be abbreviated, not listed in full. The year of publication should be listed at the end of the reference, not after the list of authors.
48) P. 18, line 4: Is this a journal article, technical report or book? Please provide full details of the Anderson (1976) reference.
49) P. 18, line 14: Provide the full range of pages for this article.
50) P. 18, line 16: Add the article # for this reference.
51) P. 18, line 26: Provide the full range of pages for this article.
52) P. 20, lines 8/9: Why is the journal name in italics?
53) P. 20, line 11: Is the French name of the journal needed here?
54) P. 21, line 6: Provide the full range of pages for this article.
55) P. 21, line 8: There is a period missing after "design".

56) P. 21, line 13: Provide the full range of pages for this article.
57) P. 21, line 31: Is there an appropriate issue number (other than zero) for this article?
58) P. 22, line 9: Provide the range of pages for this article.
59) P. 22, line 19: Use upper case "H" in "Journal of Hydrology".
60) P. 23, line 16: Why is this "Woo et al. (2008a)" when there is no corresponding "Woo et al. (2008b)"?
61) P. 24, Figure 1: I presume the upper and lower divisions shown in each catchment are delineated by the black contours? If so, the figure caption should clearly state this. The range of colors is misleading since there does not appear to be elevations above 1000 m. As such, consider using a shorter range of elevations for the map with more distinctive colors.
62) P. 25, Figure 2: For which year(s) are these results valid for? Is this for a given year or a climatology over the study period?
63) P. 26, Figure 3: Here snow cover extent is expressed as a percentage in the color legend but in Figure 2 it was shown as a fraction from 0 to 1. Use a consistent parameter for the presentation of the results. The range of elevations for each zone should be provided in a table.
64) P. 27, line 27: The date format may need revisions.
65) P. 28, line 34: Same comment.
66) P. 29, Figure 6: Convert the SWE data from inches to mm and redraft the figures accordingly.
67) P. 30, Figure 7: Provide units for RMSE on the y-axis. Would it be possible to have ovals around the different clusters to identify specific basins on the plot?
68) P. 31, line 51: Change to "on the plots".
69) P. 32, Figure 9: Discharge should be in units of $m^3/s$ on the y-axis. Rather than presenting the average results over 10 water years, why not depict results for each ablation season?
70) P. 33, Table 1: For the upper Little Chena, provide the air temperature with one decimal, i.e. "-21.0" for consistency with values reported elsewhere.
71) P. 34, Table 2: Is the average SWE reported here the annual average, or the average annual peak value?
72) P. 35, Table 3: There are a couple extraneous numbers in the table just under "Max" ("13" and "14", which appear to be line numbers. The maximum MBASE temperature should read "0.00".
73) P. 36, Table 4: Probability values should be reported for the correlation statistics.

**New References:**

Déry, S. J., Salomonson, V. V., Stieglitz, M., Hall, D. K., and Appel, I. 2005: An approach to using snow areal depletion curves inferred from MODIS and its application to land surface modelling in Alaska, Hydrol. Proc, 19, 2755-2774, doi: 10.1002/hyp.5784.

Hall, D. K., Riggs, G. A., Foster, J. L., Kumar, S. V. 2010: Development and evaluation of a cloud-gap-filled MODIS daily snow-cover product, Remote Sens. Env., 114(3), 496-503.

Tong, J., Déry, S. J., and Jackson, P. L., 2009: Topographic control of snow distribution in an alpine watershed of western Canada inferred from spatially-filtered MODIS snow products, Hydrol. Earth Syst. Sci., 13, 319-326.

---

## Author Comment (AC1) · 19 Sep 2018

Using MODIS estimates of fractional snow-covered area to improve streamflow forecasts in Interior Alaska by Bennett et al. examines improvements in model skill when remotely-sensed snow-covered area estimates are used to model streamflow, compared to model runs where model-generated areal depletion curves are used. For this study, two MODIS-derived snow covered area products were used, MOD10A1 and MODSCAG. This study is a nice assessment of the use of remotely-sensed snow cover products with the new CHPS modeling framework for several watersheds in the interior of Alaska. This study demonstrates the improvements as well as pitfalls of using areal depletion curves vs. remotely sensed snow-covered area. The authors find that using remotely sensed snow-covered area yields modest improvements in some basins, especially the sparsely measured ones, but not in others.

These findings agree with previous studies, which the authors cite. Overall, the techniques are well researched and the findings are sound, but I have a few major concerns that I would like to see addressed prior to publication:

Thank you for your review and your positive words about our study.

1) In most of the cited publications, e.g. Painter et al. (2009); Rittger et al. (2013), what is referred to in this manuscript as snow covered extent is called fractional snow-covered area, or fSCA. Since MODSCAG and MOD10A1 are both fractional products, fractional snow-covered area is a more accurate term than snow-covered extent. Thus, I suggest changing snow-covered extent to fractional snow-covered area to align with most other publications.

We agree with this point, and we have changed the referral of fractional snow cover extent (SCE) to fractional snow covered area (fSCA) when we are discussing the remotely sensed products in the paper.

2) What is really needed for model input is the total volume of snow water equivalent (SWE). The fSCA contains no information on depth. Among other problems, as the authors point out, when fSCA reaches 100%, it gives little information about the snow volume. I realize that there is no good direct SWE estimate for model input, however there have been many attempts to create basin-wide SWE estimates, for example by fusing snow telemetry estimates with fSCA (Fassnacht et al., 2003; Dozier et al., 2016; Schneider and Molotch, 2016). It would be worthwhile to at least discuss why fSCA only was chosen to improve the streamflow forecasts.

We agree with your point, and if were we working in the lower 48, our study would have been set up differently. SWE is the more important variable, and the 'grail' for water resources managers and snow scientists. However, in Alaska, there are limitations with regards to the ground-based observations to carry out validation and testing of basin-wide SWE simulations and the remote sensing of snow and topography that can be used for simulations in this environment. Regarding a), ground-based observations such as SNOTEL, as used by Fassnacht et al. (2003) and Schneider and Molotch (2016), are available in Alaska. However, as noted by Fassnacht et al. (2003), to interpolate between the stations there should be a minimum distance between them. In the Upper Colorado River basin (277,000 km$^2$) there are noted to be 240 SNOTEL sites, operating since 1991. In Alaska, there are 40 SNOTEL sites (1.718 million km²), and in the basins where we undertook this study (10,160 km$^2$) there are 7 SNOTEL stations, all of which are located in the Chena River basin. Another issue is the quality of SNOTEL data, including station siting, have been noted by various authors, although the

Alaska SNOTEL station network is not included in these reviews (Dozier et al. 2016, Oyler et al. 2015, Ragwala et al. 2015).

Issues with remote sensing in Alaska are related to availability and quality of polar orbiting remote sensing products, availability of data on snow pack depth and snow density (Muskett 2012), issues related to deep and shallow snow packs, issues related to the mapping forests cover fractions and the density of boreal forest canopies. Unfortunately, many of the remote sensing of snow products that are available are not tested well in the high latitudes and under dense boreal forest cover, which highlights the importance of our work in these regions and necessitates a simpler initial approach to research in the region.

Additionally, the availability of digital elevation models (DEM) information in Alaska hinders the kind of analysis performed in the aforementioned studies; Alaska's 5-m IFSAR product is nearing completion in 2018 but was not available when this project was carried out. The 30 m National Elevation Model (NED) DEM used in the study likely contains issues (that will be corrected by the updates to the IFSAR Alaskan product), including data voids, data currency, geodetic datum issues. For example, all datums for Alaska DEMs were previously in NAD27, which caused offsets in the data (Maune, 2008). For these reasons, it has previously been difficult to successfully apply regression or interpolation of fSCA to extract SWE. We are hopeful with the IFSAR DEM for Alaska, these methods may be applied for future work.

3) The interpolation, filtering, and smoothing of both MOD10A1 and MODSCAG is barely mentioned in the text and the supplement. Snow-cloud discrimination and how MODIS data are smoothed is a critical step that the authors have, at the least, not fully addressed. Likewise, viewing geometry also greatly affects the accuracy of MODIS surface reflectance (Tan et al., 2006), which both MOD10A1 and MODSCAG are based on. I recommend the following two studies as examples of different smoothing approaches for snow cover from MODIS, Dozier et al. (2008); Morriss et al. (2016). I would like to know how the authors' approach compares to these two smoothing techniques.

Our interpolation, filtering and smoothing of MODIS data is dealt with through pre-processing and in the CHPS software. We have added more detail through the paper, and the supplement, to reflect this question. In addition, we have added these references to the paper.

Interpolation: We used the MODIS Re-projection Tool (Dwyer and Schmidt, 2006) to pre-process imagery into an Alaska Equal Area Conic projected GeoTIFF of fractional SCA (USGS, 2011). This preprocessing step assisted us to correct, in part, the viewing geometry and other issues related to projections of the original MODIS data and the influence these projections have on the MODIS data for Alaska. We interpolated the data using Nearest Neighbor interpolation methods available in this tool. We interpreted only cloud-free pixels.

Filtering: We input the MODIS data products, with corrections for viewable area. While we did experiment with a cloud correction, and also with different sized aggregates of MODIS grid cells to determine the influence of spatial aggregation approaches, we only applied the tree correction as detailed in the paper. The reasoning for this is that these methods did not make a difference within the CHPS software, while only considering streamflow responses. We discuss in the paper why we might want to look at different metrics to really evaluate these types of pre-processing methods.

Smoothing: We ingested the MODIS data into CHPS. CHPS provided several different means to filter and smooth the data. First of all, there is the optional element in CHPS, maxGapLength, can be configured to define the maximum length of gaps that should be filled. Gaps equal to or smaller than maxGapLength will be filled with interpolated values, while gaps larger than

maxGapLength will not be filled. This ensures that periods with extensive cloud cover obscuring the MODIS fSCA data are interpolated but long periods with no data, such as the summer period, are not interpolated. A maxGapLength of 11 days was selected after testing revealed that longer and shorter interpolation time steps resulted in lower streamflow simulation skill. We describe the use of maxGapLength in the supplemental.

I have included minor comments as an annotated PDF. Several citations in the text were not in the bibliography. Thus, I suggest the authors carefully check that the citations in the text correspond to those in the bibliography. If the authors have any questions about my review, I encourage them to contact me at nbair@eri.ucsb.edu.

Thank you very much. We have gone through and addressed each point you have included as minor comments. They are listed below in incremental order. We have also gone through the paper and addressed all of the comments in the annotated PDF document provided. Please see the attached track-changes version of the manuscript as well.

1. We added to the sentence Two versions of MODIS fSCA are tested "against a base case aerial depletion curve-derived extent of snow cover".
2. We changed this to "a myriad of impacts".
3. We added the SWIPA report to the bibliography and also added several more references on snow melt disappearance timing. Although there are challenges, as the reviewer notes, snow covered area estimates and snow melt timing and disappearance timing are considered more robust. On the other hand, SWE estimates, snow depths, and snow density are elusive measurements that have high spatial variability and are not easy to obtain in Alaska, and globally.
4. We deleted the sentence and added Huntington into the bibliography. We added the Cohen reference to the previous sentence.
5. We deleted the second occurrence of "model output" on page 3 of the manuscript. Thank you for catching this.
6. In the equation, we changed *e* to *elev*.
7. Address the SB constant, look at UADJ. This was a confusing sentence, and we re-wrote it to make it clearer what the assumptions of SNOW17 are. Hopefully this clears up the two issues that you had raised.
8. We deleted gradient from the sentence.
9. We added units to Figure 1.
10. Regarding the low bias in the SWE estimates, we are comparing station locations to the modeled results for the entire Upper Chena River basin (lumped, north and south units are shown separately) unit. Thus, we would expect the average across the lumped basin unit to be lower than the SNOTEL sites. This is accentuated in some sites more than others, for example there is a lot more snow at the Upper Chena SNOTEL gage in water year 2001, as opposed to the Teuchet site where less snow fell. We added some additional clarification where we discuss the results of this figure in section 3.3 of the revised paper.

Sincerely,
Ned Bair

Earth Research Institute
University of California, Santa Barbara

References:

Dozier, J., Bair, E.H. and Davis, R.E., 2016. Estimating the spatial distribution of snow water equivalent in the world's mountains. WIREs Water, 3: 461-474.

Dozier, J., Painter, T.H., Rittger, K. and Frew, J.E., 2008. Time-space continuity of daily maps of fractional snow cover and albedo from MODIS. Advances in Water Resources, 31: 1515-1526.

Fassnacht, S.R., Dressler, K.A. and Bales, R.C., 2003. Snow water equivalent interpolation for the Colorado River Basin from snow telemetry (SNOTEL) data. Water Resources Research, 39(8): 1208.

Morriss, B.F., Ochs, E., Deeb, E.J., Newman, S.D., Daly, S.F. and Gagnon, J.J., 2016. Persistence-based temporal filtering for MODIS snow products. Remote Sensing of Environment, 175: 130-137.

Painter, T.H., Rittger, K., McKenzie, C., Slaughter, P., Davis, R.E. and Dozier, J., 2009. Retrieval of subpixel snow-covered area, grain size, and albedo from MODIS. Remote Sensing of Environment, 113: 868-879.

Rittger, K., Painter, T.H. and Dozier, J., 2013. Assessment of methods for mapping snow cover from MODIS. Advances in Water Resources, 51(1): 367-380.

Schneider, D. and Molotch, N.P., 2016. Real-time estimation of snow water equivalent in the Upper Colorado River Basin using MODIS-based SWE reconstructions and SNOTEL data. Water Resources Research, 52(10): 7892-7910.

Tan, B., Woodcock, C.E., Hu, J., Zhang, P., Ozdogan, M., Huang, D., Yang, W., Knyazikhin, Y. and Myneni, R.B., 2006. The impact of gridding artifacts on the local spatial properties of MODIS data: Implications for validation, compositing, and band-to-band registration across resolutions. Remote Sensing of Environment, 105(2): 98-114.

References:

Rangwala, I., Bardsley, T., Pescinski, M., and J. Miller (2015). SNOTEL sensor upgrade has caused temperature record inhomogeneities for the Intermountain West: Implications for climate change impact assessments. Western Water Assessment Climate Research Briefing.

Maune, D.F. 2008. Digital Elevation Model (DEM) Data for the Alaska Statewide Digital Mapping Initiative (SDMI). A report prepared for Alaska Geographic Data Committee. 161 pp.

Maune, D.F. 2009. Alaska Statewide Digitial Mapping Initiative, Mapping Alaska for the First Time.

Muskett, R.R., 2012. Remote sensing, model-derived and ground measurements of snow water equivalent and snow density in Alaska. *International Journal of Geosciences*, *3*(05), p.1127.

USGS, 2011. MODIS Reprojection Tool User's Manual. Land Processes DAAC  USGS Earth Resources Observation and Science (EROS) Center. Accessed August 15, 2018. https://lpdaac.usgs.gov/sites/default/files/public/mrt41_usermanual_032811.pdf. 69 pp.

---

## Author Comment (AC2) · 19 Sep 2018

In their study "Using MODIS estimates of fractional snow cover extent to improve streamflow forecasts in Interior Alaska" Bennett et al. investigate the value of two MODIS-derived snow cover area products (MOD10A1 and MODSCAG) to improve streamflow simulations in the interior of Alaska as compared to simulations where model-generated areal depletion curves are used. The authors conclude that there is only marginal improvement when evaluating the model performance with metrics such as NSE, RMSE etc., but argue that the MODIS derived snow covered area products might be valuable particularly in regions with sparse or poor quality observations. The methods and findings are sound and the article is generally written in a clear, concise and very structured way.

Thank you for your positive words about our study.

Nevertheless, I have some minor comments, which I would like to see addressed prior to publication:

General comments
• The basins with the sparsest streamflow observation had the greatest improvement in streamflow simulations. (P1L13,P14L 25-26) –> could indeed the sparse data be the main reason for the improvement rather than the product? This could be tested with a little model experiment adding some data gaps and see how the performance measure is sensitive to that maybe for the basin with the longest observations.
We are unsure if we could simply remove missing data from a 'good' record to test this. It is difficult to pull apart whether improvement is due to the missing streamflow observations, or if it is due to improving the snow observations in the catchment. However, it makes logical sense that improving observations (whether snow or streamflow) would be beneficial when before there are no observations. We talk about this in the discussion section of the paper. If this is a sticking point for the reviewer, we can put some thought into how to best test this, and perform the experiment for one watershed, and one year, as an example.

• Why do the authors include the Chatanika catchment, when it shows very poor performance both using the model generated and the MODIS product derived snow cover extend? Please clarify.
We were interested to show the results for the Chatanika as we feel that this basin is most representative of the poor-quality streamflow observations in Alaska. We think that even though it didn't perform well, that this is likely the kind of issues modelers and observationalists will deal with when working with Alaskan hydrology data. This basin is also adjacent to the Caribou-Poker Creek instrumented watershed, and it represents a lowland site in comparison to the Salcha, Chena and Goodpaster systems which have more upland land cover. For this reason, we would like to keep the basin included in the paper, despite its poor performance.

• Move more detailed description of the derivation and differences between the MODIs products (interpolation, filtering, and smoothing) from the supplements in the main study.
We have now moved more of the details from the MODIS productions to the body of the paper. We originally took it out because we felt that the Methods section was too long, and we didn't want to bog readers down. Based on this comment, and comments from Reviewer 1, we were too gregarious with these edits.

Minor/technical comments
• There are passages in the article where fill words like utmost, great, very are used abundantly. In my opinion, it would help to go through the article and check where these are really needed and where they could be dropped.

We have dropped these words throughout the text. Please see the track changed version of the manuscript.

• There is a mixed use of watershed and basin. Do the authors use it with equivalent meaning? If so, then use only one term throughout the paper, if not please clarify why the two terms are used.
We have removed the use of the word watershed and replaced it with basin throughout the paper. Thank you for noting this.

• Often model runs could be replaced for better clarity with SWE simulations (e.g. P10L30) or streamflow simulations, respectively.
We have replaced run(s) with simulation(s) throughout the paper.

• Please make the units consistent (sometimes there is sec, sometimes there is s, etc.)
We have made the units consistent throughout the text. Please see the track changed version of the manuscript.

P1L29/30 did both extent and duration decrease by the same percentage?
We have adjusted this sentence to read "Snowpack extents in Alaska have decreased over time by 18% (1966-2012) due to an earlier snow melt, while snowpack duration has also decreased (SWIPA, 2012)."

P1L35 Delete this sentence
We changed this sentence to read as follows "Rivers in Alaska have been observed to be changing as a result of an intensified or stronger hydrologic cycle that could lead to an increase in peak flows in the Northern American high latitudes (Cohen et al. 2012; Huntington, 2006; Rawlins et al., 2010)."

P2L3 Extremes –> Extreme
We corrected this typographic error. Thank you for noting it.

P3L14 in which P3L22/23 what does it mean that they perform better, better than in other regions or these models are better in these regions than the other models. Please clarify.
We meant that temperature index models are presumed to perform better than other models in highly variable landscapes with spare networks. We have added in "…than other models for regions…" to the sentence to clarify.

P3L33 Date missing in reference
We have added the date to the reference. Thank you.

P4L33 delete above the Steese Highway (I do not see the relevance of this information)
We have deleted this part of the sentence.

P5L23 delete at the Steese Highway site
We have deleted this part of the sentence.

P5L35 add eq. 1 in brackets
We have added Equation 1 to the first equation in the paper ($SCA_{fadj}$), and Equation 2 for this equation.

P6L1 does still really need to be expressed in feet?
Because of the way that the equation was developed, you cannot obtain the same answer by converting 1000 ft to meters. Thus, I believe that you must enter the value of the elevation in feet. It is confusing because I used meters in the example. I have now added the meters to feet conversion to make it clearer, and added the unit value after elevation in brackets.

P6L17 how sensitive are streamflow simulations to this lapse rate? What motivates the assumption that the fixed lapse rate of 0.6_C/100m holds?
This lapse rate is a default in SNOW17 to represent the saturated adiabatic lapse rate, and is used to calculate the percentage of the watershed where precipitation falls as snow. However, the value can be changed for each basin and sub-basin, if warranted by the input temperature data, and also different methods can be applied to separate rain versus snow. The lapse rate is used to find the air temperature threshold value, and this value is used to relate to an elevation, for which the basin area snow fall can be calculated. It is important to note that this is different from other uses of lapse rates in the model.
Although we could not find previous studies that account for the sensitivity of this parameter, there are six main parameters in SNOW17 that have been identified as the most sensitive parameters for SNOW17, SCF, MFMAX, MFMIN, SI and UADJ (Anderson 2002, Tang et al. 2007). The use of the single lapse rate value for these calculations is widely applied in studies across the globe (e.g. Clark et al. 2011). We have added more detail and these references to this section of the manuscript.

P6L30 delete "and is set to . . . "already mentioned above
We have deleted this part of the sentence.

P6L33 mm/mb/6hr is that the unit for rain on snow, then move to melt from rain of snow earlier in the sentence
We have moved mm/mb/6hr to come after UADJ events in the revised sentence.

P7L12-15 It might be helpful to see a sketch of how the ADC works.
While we agree it might be useful, we have many figures in this paper already. Thus, we have added a reference to the images that depict the ADC relationships in Anderson 2002's paper (Anderson 2002, Figure 7.4.3, 7.4.4).

P7L16 add a reference to this look up table
We have added in the reference for the ADC look up table.

P7L29 change "produces streamflow simulates" to "simulates streamflow"
We have changed this sentence as suggested.

P8L12 delete study
We have deleted study.

P8L30 to P910 all of these statistics are well known, I think it would be sufficient to just add some reference for each, else, I would summarize them in a separate table. In case the authors want to keep the equations, check the equations carefully: MAE has to additionally be divided by the number of observations, Spearman correlation coefficient could simply be written in a simpler form.
We agree with this comment and have opted to remove the section and add references for a few of the statistics.

P9L10 make units consistent
We are unsure which units you are referring to here, but we have reviewed all units in the manuscript to ensure they are consistent. Please let us know if we have missed anything.

P10L1 delete "for reasons that are discussed in the following section"
We have deleted this part of the sentence.

P10L5 why it is the maximum recommended value and who recommended it? Maybe refer again to the table with parameter ranges.
Anderson 2002 recommends the ranges for these parameters. We have changed the sentence to include the reference and refer to the table.

P10L14 What does a more rigorous calibration mean here?
We have deleted this sentence.

P11L10 Why for May 15th 2001?
The May 15th date for this region in Alaska represents a time when snow is melting, and we should be partway through the snow recession. For this reason, some snow will be represented at higher elevations and likely less at lower elevations. The year, 2001, was selected somewhat arbitrarily, it is a moderate snow year, which we thought would show these relationships and differences across MODIS data more clearly.

P12L6 1-R
We have corrected this typographical error. Thank you.

Figures
- Most figures are difficult to read when printed in black and white. This could be improved easily by adapting the color palette.
  We changed the color palettes in most of the figures.
- Figure1: not all elevation classes are used in the map. Units are missing for the elevation zones. Drop last sentence in caption.
  Although hard to see, the upper two elevation classes (green shades) are found in the Salcha and Goodpaster basins. However, we have revised the color palatte and color ramp on the figure.We added elevation units and dropped the last sentence as suggested.

- Figure2: what happens to MODIS SASC North between May 10 and 17?
  It looks like the MODIS SASC recorded that snow cover extent increased to 80% of the Upper Chena River basin on May 13th, 2010 at 6:00 am. After this point, at the next recorded interval the snow had all melted. We correlated this with the SNOTEL gauge across the Upper Chena river basin, which are Upper Chena, Teuchet Creek, and Monument Creek. Although it looks like some precipitation fell on the 11th, no snow fell at all around this time. Thus, we believe this is an error in the MODIS data, potentially where clouds were interpreted as snow cover. Also, comparing directly with the SNOTEL gages indicates that all snow cover extents should also have been at zero at this point, however all model results indicate that there was still residual amounts of snow (0.1 fractional SASC) in the catchment. However, the plot is meant to show the differences between the SASC in the SNOW17 when different remote sensing tools are applied. Therefore, to not distract the reader, we opted to remove this outlier data point as quality control. We have not changed the text as we feel that this level of detail is not warranted, but if you think we should explain the removal of the data point, we will add in a sentence.

- Figure3: It seems there is a difference in the fractional snow cover extend seasonal development for the years that are used for calibration and the years that are used for validation. Is this also the case for each individual catchment, or is one catchment causing this difference? Is it ok to shift the year in validation, calibration period in the Chatanika catchment compared to the other catchments?
This figure shows the average snow cover extent across all the catchments based on elevation bands, and we tried to capture high and low streamflow years in the calibration and validation periods. Although, the figure shows the variability in snow cover extent across the years, it also shows that there are high and low years where there was variability across elevation zones in the melt trajectories, some years where there was a larger range of snow melt out dates (2000, 2006), while other years there was more consistency in the melt out (2002, 2007). We shifted the calibration and validation period in the Chatanika due to the improvement in quality of the data for the last 5 years of the record, and we think this is ok. To show the variations in the streamflow data, we show all the years for the Upper Chena (Figure 1) and the Chatanika basins (Figure 2) are given below.

[Figure]

*Figure 1. The Upper Chena River basin observed streamflow for the calibration and validation years.*

[Figure]

*Figure 2 The Chatanika River basin observed streamflow for the calibration and validation years.*

To show the variations in the catchments we generated the year 2000 based on the upper basin areas (Figure 3). We do not think there is a lot of variation across the basins, and hence we feel that the variability observed in each panel in Figure 3 is due to the elevation differences, and the year-to-year variations in climate, which occur across all basins.

[Figure]

*Figure 3. Four upper basins and their SCA (%) values.*

- Figure9: could the time be extended spanning April to September as also used in Table 1? Would it be possible to add quantiles of the streamflow to get an idea about the range per season? Unit should be m3/s, I do not understand what the average of all basins can tell me.
  We decided to change the table to align with Figure 9. So, the table now shows precipitation values for Nov-Dec-Jan-Feb (winter), and Mar-Apr-May-June (spring). Because we calibrated the models only for the snow melt season, the rainy season is not illustrated in the plots. Also, we corrected the units in the plots. The average of all basins was included to show the regional streamflow magnitude and hydrograph shape for the average of all basins. We have opted to retain it in the plots but if you feel strongly about this we can remove it.

Tables
- Table1: units: elevation m a.s.l.; Q m3/s?
  We have corrected the Table, and changed the period of P to align with Figure 9.
- Table2: maybe replace current with the last year included.
  We have replaced current with last year as suggested.
- Table3: SCF Max values seem messed up
  We are unsure what you mean by this comment. The SCF values ranges from 0.65 to 0.95 across the catchments.
- Table4: mention that period of record is not the same for each catchment!
  We have added this to the table caption as suggested.

References:

Anderson, E.A., 2002. Calibration of conceptual hydrologic models for use in river forecasting. Office of Hydrologic Development, US National Weather Service, Silver Spring, MD. http://140.90.113.200/oh/hrl/modelcalibration/1.%20Calibration%20Process/1_Anderson_CalbManual.pdf Accessed August 17th, 2018. pp 372.

Clark, M.P., Hendrikx, J., Slater, A.G., Kavetski, D., Anderson, B., Cullen, N.J., Kerr, T., Hreinsson, E.Ö. and Woods, R.A., 2011. Representing spatial variability of snow water equivalent in hydrologic and land-surface models: A review. *Water Resources Research*, *47*(7).

Tang, T., P. Reed, T. Wagener, K. Van Werkhoven. Comparing sensitivity analysis methods to advance lumped watershed model identification and evaluation. Hydrology and Earth System Sciences Discussions, European Geosciences Union, 2007, 11 (2), pp.793-817.

---

## Author Comment (AC3) · 19 Sep 2018

In this paper, the authors employ daily Moderate Resolution Spectroradiometer (MODIS) fractional snow cover extent (SCE) data to improve streamflow simulations in several Alaskan sub-watersheds of the Tanana River. The study period covers 2000-2010 with simulations with the SAC-SMA conceptual rainfall-runoff model that also incorporates the one-layer SNOW17 model for the representation of snowpack conditions. Runoff simulations that include MODIS-derived snow areal depletion curves (ADCs) in SNOW17 are compared with baseline simulations with the standard model formulation for ADCs in the five sub-basins of the Tanana River. The authors conclude that the assimilation of the MODIS SCE data leads to better representation of snow conditions and runoff simulations in Interior Alaska.

This paper presents interesting results on the potential application of MODIS SCE data in operational models for improved runoff simulations in Interior Alaskan watersheds where in situ data remain sparse. The paper is generally well-written and illustrated, but the paper requires some revisions prior to publication.

Thank you for your positive words and your detailed review. We feel that your suggestions, along with other reviewers, have vastly improved the paper.

The following provides a list of suggestions that may be helpful to the authors in revising their paper:

General Comments:
1)  The paper includes non-metric units including feet for elevations and inches for snow water equivalent (SWE). Please convert all non-metric units to metric and adjust Equation (1) accordingly. We added units to the elevation map (m). We have changed the inches to mm as referred to in the text and shown in Figure 6.
2)  A considerable amount of effort has been placed into ingesting the MODIS SCE data into SAC-SMA model simulations of runoff in five sub-watersheds of the Tanana River. The authors should be commended for this effort. Nonetheless, the results shown in Figure 9 show little differences between the simulations that incorporate the MODIS data versus those with the standard model formulation. Table 4 confirms there are only very modest gains to be made by ingesting the MODIS data into the runoff simulations. As stated by the authors, more significant gains would be obtained by having more accurate forcing data (air temperature and precipitation) in the remote and complex terrain of these Alaskan watersheds. Further to this, SNOW17 incorporates only one snow layer which may miss some of the snow dynamics at play within the thin snowpack layer than interacts with the atmosphere. As such, why spend so much effort in trying to improve the runoff simulations with the data assimilation strategy when more significant gains may be obtained by improving other aspects of the modeling framework? This is an interesting comment, and it highlights a point regarding the work that isn't necessarily raised in the paper. We went to considerable effort to ingest MODIS data into the modeling framework that is being used operationally in Alaska by the Alaska Pacific River Forecast Center (APRFC). We wanted very much to work closely with the APRFC on the effort, so that our work could feedback to their operational workflow. There was a lot of interest within the APRFC in ingesting remote sensing tools, and this study pointed out that there are some gains to be made, but other efforts (e.g. improved climate station data, and model ensembles, including physical models) should be pursued as well. And, it also points to the need for a more flexible calibration scheme that considers all available ground based and remotely sensed observations, including SWE, fSCA, in addition to streamflow observations. Considering that we have received support to implement the operational stages of this work, and also test a

physically based model in the state, we feel that this effort was an important step and was valued by the stakeholders involved in the project.

3) Why does the study period cover only 2000-2010 when MODIS data are available up to present? Further to this, how are gaps in the MODIS data in-filled? For instance, persistent cloud cover can lead to a significant reduction in the available snowcover data from optical remote sensing. Is any gap-filling procedure used to address this issue (see for example Hall et al., 2010 and Tong et al., 2009). The study period reflects the time when the work was undertaken. Although there is more recent information, we feel that the 2000-2010 time period is sufficient to illustrate our points, and we do not think the message of the paper would change by adding more data to the study. Regarding the infilling of gaps, we have now added more details regarding how we pre-processed the MODIS data outside of CHPS, and within CHPS, in section 2.2 of the paper.

4) Hydrological simulations such as those presented in Figure 9 are averaged over 10 water years. Results for each individual year should also be presented to illustrate the model's ability to represent interannual variability in the discharge patterns. We have generated the figures for each year, and we have included these in the Supplemental. We refer to them in the text ~page 15.

5) The references need to be fully revised and presented in the journal's standard format. We have revised the references as suggested.

6) Note that Déry et al. (2005) used MODIS ADCs to improve their simulations of runoff on the Alaskan North Slope and may be a relevant reference to this study. We have included a reference to Dery et al. (2005) in the paper. This was an oversight on our part, thank you for pointing this out.

Specific Comments:
1) P. 2, Abstract: Include the study period within the abstract. We have included the study period in the abstract.
2) P. 2, lines 5 and 25: Define "US". We now define US in the Abstract and Introduction.
3) P. 2, lines 29/30: Have both snow cover extent and duration in Alaska indeed declined by 18% from 1966 to 2012? We have corrected this line per Reviewer #2 comments to read "Snowpack extents in Alaska have decreased over time by 18% (1966-2012) due to an earlier snow melt, while snowpack duration has also decreased (SWIPA, 2012)."
4) P. 2, line 31: What aspect of permafrost has declined in response to warmer air temperatures in Alaska? Its depth, extent, or other characteristic? We have added thaw to this sentence.
5) P. 2, line 34: Change to "North American". We have made this change. Thank you for pointing this out.
6) P. 3, line 3: "Extremes" should be singular. We have made this change.
7) P. 3, line 16: Delete the extra "model output". We have deleted these words.
8) P. 3, line 20: Define "NOAA". We have added the definition.
9) P. 3, line 30: Change to "these data have". We have changed this to these and has to have.
10) P. 5, line 15: Why does the study period end in 2010 although MODIS data are available up to present? See answer to this question above in general comments.
11) P. 5, line 27: Define "SWE" upon first usage rather than in the following line. We have added the definition. Thank you for pointing this out.
12) P. 5, line 35: Perhaps number the equations, depending on the journal's formatting guidelines. Convert the equation to metric units and ensure the elevation $e$ is in meters, not feet.

We have added numbers for the equations. See response to Reviewer #2 comments regarding this equation.

13) P. 6, line 3: Define "SAC-SMA". SAC-SMA is defined on page 4 of the revised manuscript, in the Introduction.

14) P. 6, line 17: Should the air temperature lapse rate be 0.6°C/100 m? Insert a space in "100 m". This was an error, we have now corrected it to read 6ºC/1000 m.

15) P. 6, line 23: The journal may prefer dates in a format such as "21 December". We have changed all dates to adhere to the suggested format.

16) P. 6, line 29: Insert a space in "100 m". We have made the correction here and elsewhere in the paper.

17) P. 6, line 31: What atmospheric temperature is used to compute incoming longwave radiation with the Stefan-Boltzmann Law?
This part of the text describes the calculation for rain-on-snow in SNOW17. From Anderson (2006, pg A-5, A-6) "T is the air temperature at ground level. Such a relationship typically assumes that the temperature of the cloud base is the same as the surface air temperature during overcast conditions and that there is fairly constant relationship between surface and upper air temperatures when the sky is clear."

18) P. 6, line 32: Why assume a constant relative humidity (RH) at 90%? Is this relative to a water (and not an ice) surface even when air temperatures are subfreezing? How does RH enter the calculation of the simplified energy balance, through the latent heat flux?
This part of the text describes the calculation for rain-on-snow in SNOW17. "When it is raining, relative humidity can be assumed to be high. With a 90% relative humidity the wet bulb temperature, the assumed temperature of the rain drops, is essentially equal to the air temperature. By making these assumptions, the energy budget equation for melt can be used to compute snowmelt during periods when it is raining" (Anderson, 2006, pg 13).

19) P. 6, line 33: How can wind have units of "mm/mb/6 hr"?
UADJ is the average wind function and has units of mm/mb/6 hr (Anderson, 2006, pg 13). We are not describing wind here. We have moved the units to fall after UADJ so it is clearer.

20) P. 6, lines 34/35: Write "snowpack" as one word. We have corrected this through the paper.

21) P. 8, line 30: Revise to: "Three additional objectives" We have corrected this sentence as suggested.

22) P. 9, lines 1 through 9: Equations numbers run on two lines and are missing for the last three equations. We have removed these equations as suggested by Reviewer #2.

23) P. 9, line 10: The units should be "m3/s". We have removed the sentence to respond to a suggestion by Reviewer #2.

24) P. 9, line 17: Provide probability values for all correlation coefficients reported in the study. We feel that the correlation coefficients and other statistics provided are sufficient. If the reviewer feels that this is a sticking point, we will calculate it.

25) P. 10, line 18: What are the units for snow density, listed here only as 0.2? The values we are reporting here is not snow density, but negative melt factor, NMF, which is a coefficient used to represent the snow heat deficit. "Snow heat deficit is either negative or positive; the rate of heat loss or gain is based on the amount of energy exchange that occurs when melt is not taking place at the snow surface." It is defined on page 9. The units for NMF mm/ºC/6 hr). See table 3 and Anderson, 2006.

26) P. 10, line 19: Insert a space in "6 hr". We have corrected this through the paper.

27) P. 10, line 35: Insert a space in "850 m". We have corrected this through the paper.

28) P. 10, line 36: Should this be "SNOW17's"? We have corrected this error. Thank you for pointing this out.

29) P. 11, lines 1 and 11: Write "snowpack" as one word. We have corrected this through the paper.

30) P. 11, line 10: Date format may need to be revised to "15 May 2001". Please also change to "is shown in Figure 5b". We have changed all dates to adhere to the suggested format.

31) P. 11, line 13: Change to "watershed's". We have adjusted this sentence. Thank you for pointing this out.

32) P. 11, lines 20 to 22: Convert SWE from inches to mm. We have changed these figures and numbers in the text to mm.

33) P. 11, line 33: Change to "improve". We have changed this as suggested.

34) P. 12, lines 4/5 and 13/15: Avoid sentences that just describe the figures – this is what figure captions are for. We have adjusted the sentences as follows: "The calibration, validation and whole period of record results shown in Figure 3, illustrates that the poorly performing basins," and we removed the sentence starting with "Here the percent…" and the sentence starting with "Plots illustrate…". We also adjusted the sentence starting with "Statistics show…".

35) P. 12, line 35: Delete "Because this." We have deleted these words.

36) P. 13, line 14: Change to "SNOW17's". We have changed this as suggested.

37) P. 13, line 17: Revise to "data are temporally". We have changed this as suggested.

38) P. 14, line 11: Write "snowpack" as one word. We have changed this as suggested through the paper.

39) P. 14, line 19: Change to "are adding". We have changed this as suggested. Thank you for pointing this out.

40) P. 14, line 20: Change to "data appear". We have changed this as suggested.

41) P. 15, line 22: Change to "have improved". We have changed this as suggested.

42) P. 16, line 11: For consistent language, change to "floods and droughts". We have changed this as suggested.

43) P. 16, line 27: Delete "to" before "during". We have changed this as suggested.

44) P. 16, lines 32 to 34: This sentence is long and confusing. Consider revising it and perhaps dividing it into two sentences. We have changed the sentences to read "The observations of rapid change in the Arctic highlight important alterations to hydrological regimes in the subarctic Interior boreal forest of Alaska. These observed, rapid changes and future anticipated alterations introduce a pressing need in Alaska to further understand the anticipated changes through modeling of major climate drivers of streamflow."

45) P. 17, line 1: Delete the space after the hyphen in "high-quality". We have corrected this.

46) P. 17, line 8: Change to "Natural Sciences". We have corrected this.

47) P. 18, line 1: Note that the references do not generally follow the format used by HESS; for instance, journal names should be abbreviated, not listed in full. The year of publication should be listed at the end of the reference, not after the list of authors. We have adjusted the references accordingly.

48) P. 18, line 4: Is this a journal article, technical report or book? Please provide full details of the Anderson (1976) reference. We have corrected this reference.

49) P. 18, line 14: Provide the full range of pages for this article. We have corrected this reference.

50) P. 18, line 16: Add the article # for this reference. We have corrected this reference.

51) P. 18, line 26: Provide the full range of pages for this article. We have corrected this reference.

52) P. 20, lines 8/9: Why is the journal name in italics? We have corrected this reference.

53) P. 20, line 11: Is the French name of the journal needed here? We have corrected this reference.

54) P. 21, line 6: Provide the full range of pages for this article. We have corrected this reference.

55) P. 21, line 8: There is a period missing after "design". We have corrected this reference.

56) P. 21, line 13: Provide the full range of pages for this article. We have corrected this reference.

57) P. 21, line 31: Is there an appropriate issue number (other than zero) for this article?

58) P. 22, line 9: Provide the range of pages for this article. We have corrected this reference.

59) P. 22, line 19: Use upper case "H" in "Journal of Hydrology". We have corrected this reference.

60) P. 23, line 16: Why is this "Woo et al. (2008a)" when there is no corresponding "Woo et al. (2008b)"? We have corrected this reference. Thank you for pointing this out.

61) P. 24, Figure 1: I presume the upper and lower divisions shown in each catchment are delineated by the black contours? If so, the figure caption should clearly state this. The range of colors is misleading since there does not appear to be elevations above 1000 m. As such, consider using a shorter range of elevations for the map with more distinctive colors. We have changed the figure's color ramp and included Elevation (m) in the legend title for the elevation zones. We have added the basin divisions to the legend.

62) P. 25, Figure 2: For which year(s) are these results valid for? Is this for a given year or a climatology over the study period? We have added the year to the figure and figure caption.

63) P. 26, Figure 3: Here snow cover extent is expressed as a percentage in the color legend but in Figure 2 it was shown as a fraction from 0 to 1. Use a consistent parameter for the presentation of the results. The range of elevations for each zone should be provided in a table. We have corrected Figure 2 to be consistent with Figure 3. The range of elevations are provided now in the Figure 3 caption.

64) P. 27, line 27: The date format may need revisions. We have changed all dates to adhere to the suggested format.

65) P. 28, line 34: Same comment. We have changed all dates to adhere to the suggested format.

66) P. 29, Figure 6: Convert the SWE data from inches to mm and redraft the figures accordingly. We have adjusted the units on these figures from inches to mm.

67) P. 30, Figure 7: Provide units for RMSE on the y-axis. Would it be possible to have ovals around the different clusters to identify specific basins on the plot? We have provided units and ovals on the figure.

68) P. 31, line 51: Change to "on the plots". We have changed this as suggested.

69) P. 32, Figure 9: Discharge should be in units of m3/s on the y-axis. Rather than presenting the average results over 10 water years, why not depict results for each ablation season? We have changed the units. We now include the 10 water years in the Supplemental.

70) P. 33, Table 1: For the upper Little Chena, provide the air temperature with one decimal, i.e. "-21.0" for consistency with values reported elsewhere. We have changed this as suggested.

71) P. 34, Table 2: Is the average SWE reported here the annual average, or the average annual peak value? We have changed the caption and the table accordingly.

72) P. 35, Table 3: There are a couple extraneous numbers in the table just under "Max" ("13" and "14", which appear to be line numbers. The maximum MBASE temperature should read "0.00". These line numbers appear in the PDF only. I will make sure they are corrected in the next stage of reviews. The MBASE temperature has been corrected.

73) P. 36, Table 4: Probability values should be reported for the correlation statistics. See response to this comment above.

New References:

Déry, S. J., Salomonson, V. V., Stieglitz, M., Hall, D. K., and Appel, I. 2005: An approach to using snow areal depletion curves inferred from MODIS and its application to land surface modelling in Alaska, Hydrol. Proc, 19, 2755-2774, doi: 10.1002/hyp.5784.

Hall, D. K., Riggs, G. A., Foster, J. L., Kumar, S. V. 2010: Development and evaluation of a cloud-gap-filled MODIS daily snow-cover product, Remote Sens. Env., 114(3), 496-503.

Tong, J., Déry, S. J., and Jackson, P. L., 2009: Topographic control of snow distribution in an alpine watershed of western Canada inferred from spatially-filtered MODIS snow products, Hydrol. Earth Syst. Sci., 13, 319-326.

Thank you for these suggestions. We have added these references to the paper.

References:
Anderson, E., 2006. Snow Accumulation and Ablation Model - SNOW-17. http://www.nws.noaa.gov/oh/hrl/nwsrfs/users_manual/part2/_pdf/22snow17.pdf, NWS NOAA, pp. 44.

---

## Referee Report (RR1)

Re-Review of "Using MODIS estimates of fractional snow cover extent to improve streamflow forecasts in Interior Alaska" by K. E. Bennett, J. E. Cherry, B. Balk and S. Lindsey

Submitted in revised form to Hydrology and Earth System Sciences

Manuscript Number: HESS-2018-96

Summary:

In this paper, the authors employ daily Moderate Resolution Spectroradiometer (MODIS) fractional snow cover extent (SCE) data to improve streamflow simulations in several Alaskan sub-watersheds of the Tanana River. The study period covers 2000-2010 with simulations with the SAC-SMA conceptual rainfall-runoff model that also incorporates the one-layer SNOW17 model for the representation of snowpack conditions. Runoff simulations that include MODIS-derived snow areal depletion curves (ADCs) in SNOW17 are compared with baseline simulations with the standard model formulation for ADCs in the five sub-basins of the Tanana River. The authors conclude that the assimilation of the MODIS SCE data leads to better representation of snow conditions and runoff simulations in Interior Alaska.

This paper presents interesting results on the potential application of MODIS SCE data in operational models for improved runoff simulations in Interior Alaskan watersheds where in situ data remain sparse. The revised paper is much improved and the authors have addressed satisfactorily the comments provided by all referees including those I submitted in my report. Prior to publication, the paper requires some additional (minor) revisions prior to publication. The following provides a list of suggestions that may be helpful to the authors in revising their paper:

General Comments:

1) The paper still includes non-metric units including feet for elevations, millibars for pressure, and inches for snow water equivalent (SWE). Please convert all remaining non-metric units to metric and adjust Equation (1) accordingly. Note that Equation (1) can be converted easily to metric units as follows:

$$C = 0.9 - [(elev - 304.8) \times 0.000353]$$

where *elev* is now in meters.

2) The journal may prefer using superscripts for all units, e.g. $m^3 \, s^{-1}$ instead of $m^3$/s. Please consult the authors' instructions on the format used for units in the journal.

3) On p.10, Section 2.5 describes the metrics used in the evaluation of the model but it is unclear at what temporal frequency the model results are for this evaluation – are these daily or monthly values? For instance, results of NSE scores tend to improve when the temporal averaging periods are monthly rather than daily. Please clarify this within the text and the appropriate figure captions.

Specific Comments:

1) P. 8, line 24: Replace "mb" with "hPa".

2) P. 12, line 10: What are the units for snow density here?

3) P. 12, line 11: Replace "northern" with "north".

4) P. 13, line 19: Change to "data have".

5) P. 13, lines 20 and 21: Why are SWE data reported in inches here?

6) P. 13, line 21: Change to "sub-basin to sub-basin".

7) P. 13, line 32: Revise to "data improve".

8) P. 13, line 33: Change to "while they perform".

9) P. 14, line 5: Change to "results shown in Figure 3 illustrate".

10) P. 14, line 13: Change to "discharge plotted".

11) P. 16, line 9: Change to "as noted in results".

12) P. 21, line 21: The "Clark et al." reference is not in the proper alphabetical order.

13) P. 22, line 13: Abbreviate the journal name.

14) P. 23, line 4: Change to "reprojection tool".

15) P. 25, line 15: Abbreviate the journal name.

16) P. 27, line 8: Insert a space in "G.: Processes".

17) P. 28, lines 13-14: The article title has all upper case letters.

18) P. 34, Table 4: What temporal frequency is used to compute the error statistics and model evaluation? Please include the probability values for the correlation coefficients, or at least denote those that are statistically-significant at a certain p-value.

19) Figure 7: At what temporal frequency are the correlation values and RMSEs computed?

20) Figure 9: The length of the x-axes should match that of the y-axes to better interpret day-to-day variations in the simulated discharge for the six study basins. Consider also changing the presentation of the results from units of discharge ($m^3 \, s^{-1}$) to specific discharge ($m^3 \, s^{-1} \, km^{-2}$) to allow easier comparisons between the basins of different areas, while keeping the range of the y-axis values identical on all six panels.

21) Supplemental, p. 2, lines 22-23: Insert spaces in "1000 m" and "1200 m".

---

## Author Response (AR2)

Reviewer #1

Thank you for completing a second, detailed review of this paper.

1. We corrected Equation 1 as suggested. Thank you. We also corrected mb to Pa in one place in the paper. However, we struggled to see where SWE was in inches, or elevation was in feet in the paper (other than Equation 1).
2. We corrected $m^3/s$ to $m^3s^{-1}$. I also changed all other slashes in units to $^{-1}$.
3. We calibrated the model in the CHPS software, which generates monthly statistics. We validated the model using monthly statistics as well. Thus, values we report in the paper are from aggregated monthly model results. We have added this to the paragraph and the figures.

Specific Comments
1. We replaced mb with hPa.
2. We added the units for snow density.
3. We replaced northern with north.
4. We changed data has to data have.
5. This was an oversight, the results are reported in mm, not inches. We correct the paper. Thank you for noting this.
6. We changed to sub-basin to sub-basin.
7. We fixed this issue.
8. We changed this to while they perform.
9. We changed this to "results shown in Figure 3 illustrate"
10. We corrected this problem.
11. We changed this as suggested.
12. We moved the reference to the correct place.
13. We abbreviated the journal name.
14. We corrected this.
15. We abbreviated the name.
16. We added the space.
17. We corrected this.
18. In the Table 4 caption we note the monthly time frequency. For R values, we italicized statistically significant values and added a sentence to the caption explaining this.
19. We added in monthly to the Figure caption.
20. For Figure 9, we changed the length of x-axes to better match the y-axis length for improved day-to-day variations in the simulated discharge for the six study basins. We also changed the results to specific discharge as suggested to allow easier comparisons between the basins of different areas.
21. We made these changes to the supplemental.

Reviewer #2

P2L9 change "aerial depletion curve-derived extent of snow cover" to "extent of snow cover derived by aerial depletion curves"
We made this correction.

P2L17 "is of value --> "valuable "
We have changed this to valuable.

P2L22 consider using another word for "tuned"
We replaced tuned with corrected.

P6L6 nearest neighbor – no capitals
We changed this to lower case.

P6L24-L29 equation 1 (not 2);please refer to the equation and do not embed it in the sentence. For instance like this: delete "The standard division by viewable gap fraction," SCAfadj (henceforth referred to simply as fSCA) is the fSCA adjusted for canopy cover (Equation 1). where Fveg is the tree cover percentage and SCAf is the unadjusted SCA data.
We have added in a reference and moved the sentence to above the equation.

P7L16/17 write "…as a coefficient, C (equation2)."
We have changed this as suggested.

P7L21 consider renaming "elev" to avoid confusion with "ELEV" from the SNOW17 model
We have changed this to *elevation*.

P8L6 6°/100m (not 1000!)
We believe this is correct as 6°C per 1000 m.

P8L14 hrs -->hr
We corrected this.

P14L5 delete "shown in Figure3" add (Figure 3) at the end of the sentence
We have changed this as suggested.

P14L12-14 This sentence appears to be incomplete please revise; delete "shown in"
We have revised the sentence, and we hope it reads more cleanly.

P16L21 delete "but not all"
We have deleted this part of the sentence.

P16L26 delete "or improved"
We deleted improved.

P16L30 interior
We have went through all instances of interior or Interior (Alaska) in the ms and cleaned this up.

P16L35 change "shown in the Liu study" to "in Liu et al. (2013)

We changed this as suggested.

P17L3 "is of value --> "valuable "
We changed this as suggested.

P18L6/7 Delete as shown in Figure 4, add (Figure4) at the end of the sentence
We rearranged this as suggested.

P18L32 Delete "system
We deleted system.

P19L6 Change "it" to "is"; break sentence in two
We made these changes as suggested.

P19L29 these references should have been used already in the introduction part (and maybe also only there)
We removed these references.

Table1: Please change the unit of Q to m3/s.
The caption does not read really consistent. Consider this proposal:
"Sub-basin characteristics, including name, sub-basin ID, area, elevation mean (range), average monthly temperature, T, for January (July), average seasonal total precipitation for winter (November-February) (spring (March-June)), annual average daily discharge Q, slope basin units (lower, N=north and S=south), land cover (based on majority cover values*). T, P, and Q calculated from the 2000-2010 water years."
We adjusted the caption as suggested, with a minor difference in the above suggested text.

Table 3: Please, check the values in cell SCF/Max there appear 3 values overprinted.
Is the unit of DYGM not per time since it is a rate?
We adjusted the description of DAYGM. We don't see any overprinting in our Word version, so we will assume this is a PDF conversion issue that will be cleared up in final print stages. We also adjusted the caption of the table to read "Minimum (Min) and maximum (Max) parameter values used in the model simulations. When min and max values are the same the parameter did not vary."

Table 4: Delete "(Per., 1999-2010)" in conflict with last sentence of the caption.
We left this the same, but changed the last sentence as follows "Note that the calibration and validation years are not the same for all catchments." as the period of record is the same but the calibration and validation years are not.

Table5: Add "upper (U)" to explain the "U" in the table; Change bolded to bold
We made this change to the table.

Figure 4: p5L24 open bracket before "blue"
We made this change. Thank you.